# Genomic evidence for homoploid hybrid speciation between ancestors of two different genera

Zefu Wang [1,3], Minghui Kang[1,3], Jialiang Li [1], Zhiyang Zhang [1], Yufei Wang[1], Chunlin Chen[1], Yongzhi Yang[2] & Jianquan Liu [1,2✉]

Homoploid hybrid speciation (HHS) has been increasingly recognized as occurring widely during species diversification of both plants and animals. However, previous studies on HHS have mostly focused on closely-related species while it has been rarely reported or tested between ancestors of different genera. Here, we explore the likely HHS origin of *Carpinus* sect. *Distegocarpus* between sect. *Carpinus* and *Ostrya* in the family Betulaceae. We generate a chromosome-level reference genome for *C. viminea* of sect. *Carpinus* and re-sequence genomes of 44 individuals from the genera *Carpinus* and *Ostrya*. Our integrated analyses of all genomic data suggest that sect. *Distegocarpus*, which has three species, likely originates through HHS during the early divergence between *Carpinus* and *Ostrya*. Our study highlights the likelihood of an HHS event between ancestors of the extant genera during their initial divergences, which may have led to reticulate phylogenies at higher taxonomic levels.

[1] State Key Laboratory of Bio-Resource and Eco-Environment of Ministry of Education, College of Life Sciences, Sichuan University, Chengdu, Sichuan, China. [2] State Key Laboratory of Grassland Agro-Ecosystem, College of Ecology, Lanzhou University, Lanzhou, Gansu, China. [3] These authors contributed equally: Zefu Wang, Minghui Kang. ✉email: liujq@nwipb.ac.cn

Hybridization has been considered as a possible factor to drive biodiversity evolution at both low and high taxonomic level for a long time[1]. In the recent past, homoploid hybrid speciation (HHS) is repeatedly evidenced to be an important mechanism that generates new species and increases biodiversity without any change in chromosome number[2]. According to the strict definition[3], HHS has to meet three criteria: genetic admixture from two parental lineages, distinct reproductive isolation (RI) between a stable hybrid lineage and its two parents, and RI resulting directly from a hybridization event. The genomic contributions of the two parents to the hybrid lineage may be equal when HHS arises from F1 hybrids, but unequal if it derives from backcrossing of hybrids[4,5]. Under the latter scenario, HHS is sometime confused with introgression (also called introgressive hybridization)[2]. All homoploid hybrids (from F1 and backcrossing) between distinct species can achieve partial intrinsic RI through fixing different allelic variations from one or other of the parents and sorting of genic incompatibilities[6,7]. In addition, introgression usually transfers adaptive alleles and helps the introgressed populations to colonize new niches where there is prezygotic RI distinct from that of the non-introgressed ones[8], although introgression of non-RI alleles between species is likely (Supplementary Fig. 1). Both introgressed and HHS lineages may therefore experience hybrid recombination of the RI-related loci, leading directly to prezygotic[5,8] and postzygotic RI[7]. However, HHS differs from introgression in that the former results in a stable lineage as a distinct taxonomic entity while the latter may be maintained as an intermittent and hybrid entity experiencing ongoing evolution, which may undergo further HHS or merge with one parent through repeated backcrossing[9] (Supplementary Fig. 1). This crucial distinction between HHS and introgression is supported by ~150 case studies based on population genetic data (Supplementary Data 1) showing that introgression produces genetic admixture in a few individuals or populations, rather than in all hybrid offspring as does HHS[2,5] (Supplementary Fig. 1).

Both HHS and incomplete lineage sorting (ILS) produce gene trees that are inconsistent between different loci across the genome, and this also results in it being difficult to distinguish between them[10]. However, ILS derives from random retention of ancestral polymorphisms across different contemporaneous species[10]. Using both outgroup and population genomic data, HHS and ILS can be discerned based on discordant site patterns and frequencies[9]. Another challenge in identifying an ancient HHS event is how to exclude effects of homoplasy[11], which may derive from random and/or convergent nucleotide mutations during long-term evolutionary histories[12,13]. Long indels (≥5 bp) extracted from well-assembled genomes can be used to effectively exclude such evolutionary homoplasy effects[11,12].

Most previous HHS events have been found to have occurred between closely-related species[14–16]; they have rarely been reported between ancestors of different genera during their initial and ancient divergences[17] and efficiently tested by multiple genomic data[11]. The genera *Carpinus* L. (comprised of sect. *Eucarpinus* Sarg. [=*Carpinus*] and sect. *Distegocarpus* [Sieb. et Zucc.] Sarg.) and *Ostrya* Scop. belong to the family Betulaceae (also known as the birch family)[18]. They contain a total of ~60 species of trees and shrubs[19]. In Betulaceae, nutlet bract morphologies are used to distinguish different genera[19,20]. The nutlets of *Ostrya* and sect. *Carpinus* are completely or rarely enclosed by bracts; however, those of sect. *Distegocarpus* are intermediate between them[19]. Three species are recognized within sect. *Distegocarpus*: *C. cordata* Blume, *C. fangiana* Hu, and *C. japonica* Blume[18,19,21]. Phylogenetic relationships among these higher taxonomic groups have hitherto remained unclear and highly debated[22–24], with sampled species of sect. *Distegocarpus* being identified as closely related to sect. *Carpinus* or *Ostrya*.

These phylogenetic contradictions have suggested evolutionary complexities among the three groups, resulting from introgression, ILS, or HHS. With the rapid growth of genomics, genomic data are capable of resolving such evolutionary uncertainties[9–12].

In this study, we focus on a likely ancient HHS event between ancestors of the two genera, *Carpinus* and *Ostrya*. We generate a high-quality chromosome-level reference genome for *C. viminea* Lindley (from sect. *Carpinus*) and re-sequence 44 individuals from the genera *Carpinus* and *Ostrya* (comprising ten species of sect. *Carpinus*, all three species of sect. *Distegocarpus*, and seven species of *Ostrya*) to test alternative hypotheses for the evolutionary relationships among the three groups.

## Results

**Genome features and comparative genomic analyses.** We assembled a chromosome-scale reference genome for *C. viminea* (Fig. 1a, Supplementary Figs. 2, 3, and Supplementary Tables 1–12; Supplementary Note 1) based on >110× Nanopore long reads, >150× Illumina short reads, and >170× Hi-C reads. We generated a high-continuity genome, with an assembly size of 372.7 Mb. A total of 242 contigs were anchored onto eight chromosomes, with contig and scaffold N50 of 4.3 Mb and 42.1 Mb respectively. A total of 26,621 protein-coding genes were predicted in the genome. We also improved our previously reported *Ostrya rehderiana* assembly[25] to the chromosome-level using ~140× Hi-C reads (Supplementary Fig. 4 and Supplementary Tables 13–16; Supplementary Note 2). We firstly explored the phylogenetic relationship and chromosomal evolution of *C. viminea*, *C. fangiana*[26], and *O. rehderiana* (as representatives of three lineages, sects. *Carpinus* and *Distegocarpus*, and *Ostrya*) with the other species in the Betulaceae (Supplementary Note 3). These three species formed a monophyletic clade, with *Ostryopsis davidiana*[5], *Corylus mandshurica*[27], and *Betula pendula*[28] from other three genera as the other three separate clades (Fig. 1b). The genomes of these three species were found to be highly conserved with no change in chromosome number and independent whole-genome duplication (WGD) event and few structural variations (Fig. 1c, d and Supplementary Fig. 5). However, numerous structural variations were detected between *Ostryopsis* and these three species, suggesting their distinct relationship (Fig. 1c).

**Phylogenetic analyses and highly inconsistent gene trees.** We then focused our study on three representative species (*C. viminea*, *C. fangiana*, and *O. rehderiana*) and further 44 re-sequenced individuals for ten species of sect. *Carpinus*, three species of sect. *Distegocarpus*, and seven species of *Ostrya* covering all major distributions of these three lineages (Fig. 2a and Supplementary Data 2). A total of 7468 strictly orthologous nuclear gene groups (1:1:1:1) were identified between *C. viminea*, *C. fangiana*, *O. rehderiana*, and the outgroup (*Ostryopsis*). After removing those with short aligned regions (<300 bp) or low alignment ratios (<50%), 6321 orthologous groups were used to generate a coalescent-based species tree, in which *O. rehderiana* was sister to a clade consisting of *C. viminea* and *C. fangiana* (Fig. 2b), consistent with the phylogenomic species tree based on the concatenated 4DTv sites (Fig. 1b). Population genomic data from the three lineages were then used to reconstruct their phylogeny. After assessment of the results, the *C. viminea* genome was used as the reference for producing more genetic variants (Supplementary Data 3). For each of the 44 re-sequenced samples, an average of ~11.0 Gb (>25×) clean bases were mapped to the reference genome, with an average mapping ratio of 89.3% and a > 20× mapping depth, covering >77% of the reference genome (Supplementary Data 4; Supplementary Note 5). Around 6.2 Mb high-quality biallelic SNPs and 443,792 indels were obtained

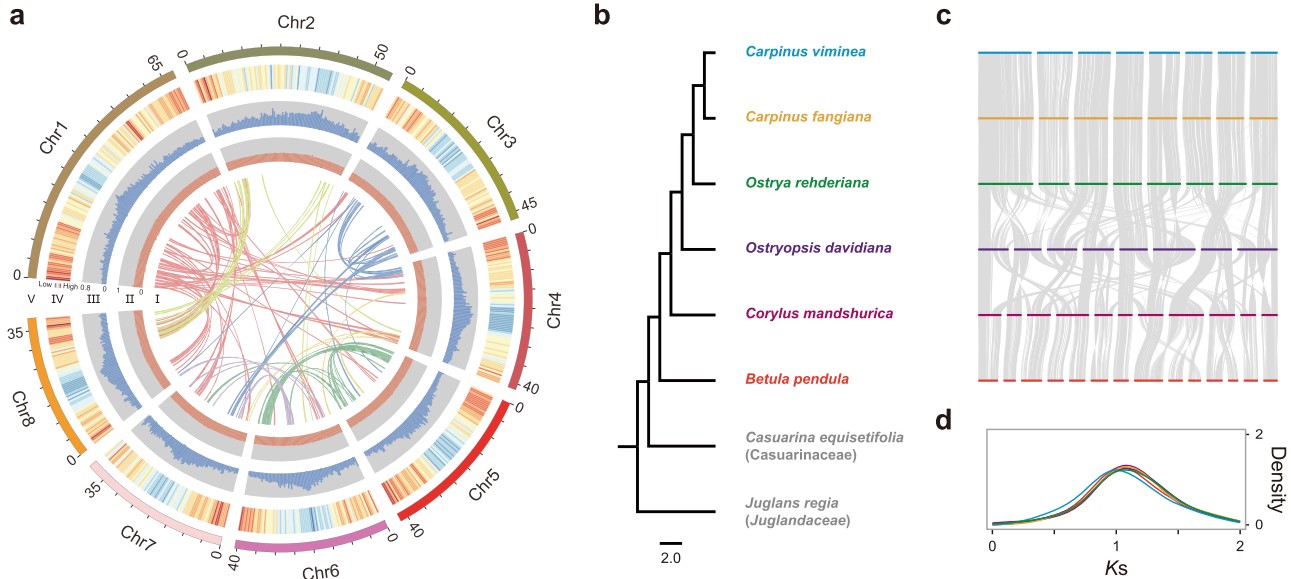

**Fig. 1 Genome features of *Carpinus viminea* and comparative genomic analyses with other species of the same family Betulaceae. a** Genome features of the *C. viminea* assembly, including the synteny information (I), GC (guanine-cytosine) content (II), repeat density (III), gene density (IV), and genome chromosomes (V). **b** Phylogenomic tree of six Betulaceae species based on the concatenated 4DTv sites. The topology was generated by RAxML, with *Casuarina equisetifolia* and *Juglans regia* as outgroups. The bootstrap values for each node were all 100. **c** MCScanX identified synteny blocks between six Betulaceae species. The chromosome colors correspond to those in (**b**). **d** *K*s distributions within each Betulaceae species. The colors of each species correspond to those in (**b**). Source data are provided as a Source Data file.

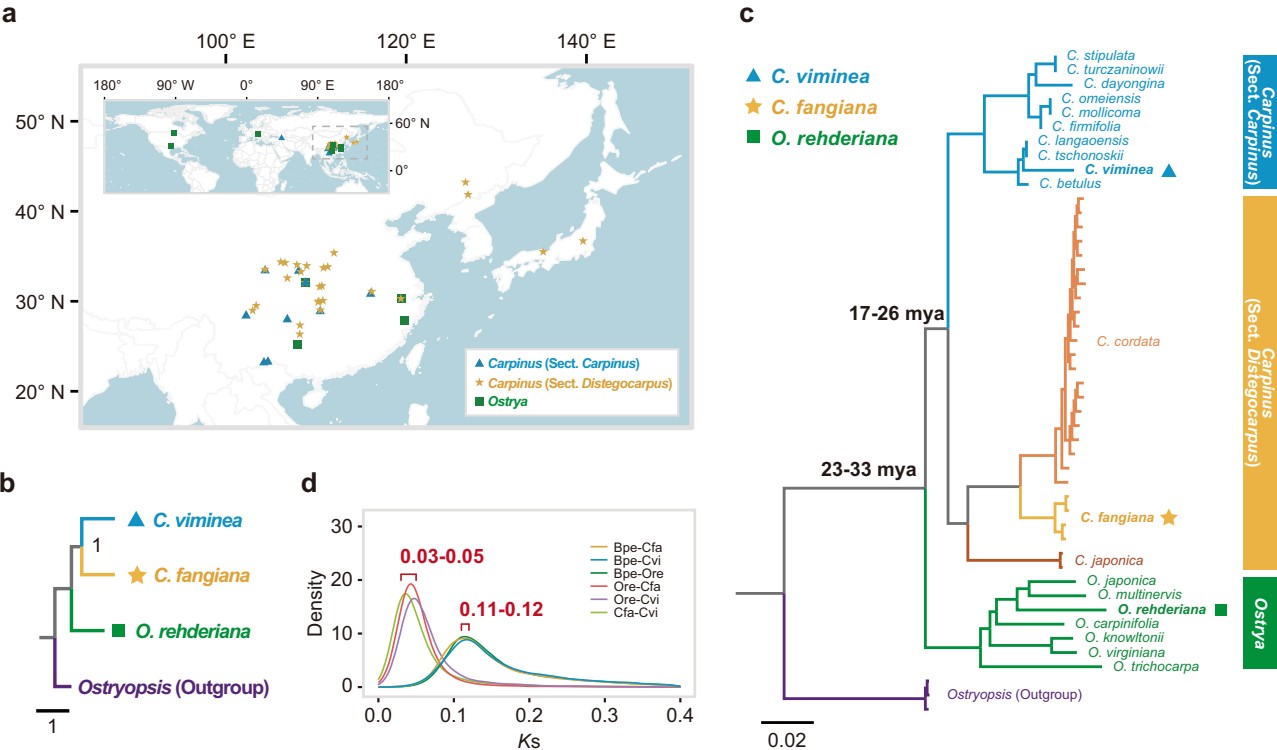

**Fig. 2 Sample locations and phylogenetic analyses. a** Geographic distributions of sampling locations for three lineages. **b** Coalescent-based species tree reconstructed using 6321 strictly orthologous nuclear gene groups by ASTRAL. The support values are estimated by local posterior probability. **c** Phylogeny of all re-sequenced samples based on analyses of nuclear SNPs. The bootstrap values for interspecific nodes were all 100. **d** *K*s distributions between members of each pair of species. *Betula pendula* (Bpe) was used as the outgroup to date the times of divergence between it and *Ostrya rehderiana* (Ore), *Carpinus viminea* (Cvi), and *C. fangiana* (Cfa). The estimated divergence times are shown in (**c**). Source data are provided as a Source Data file.

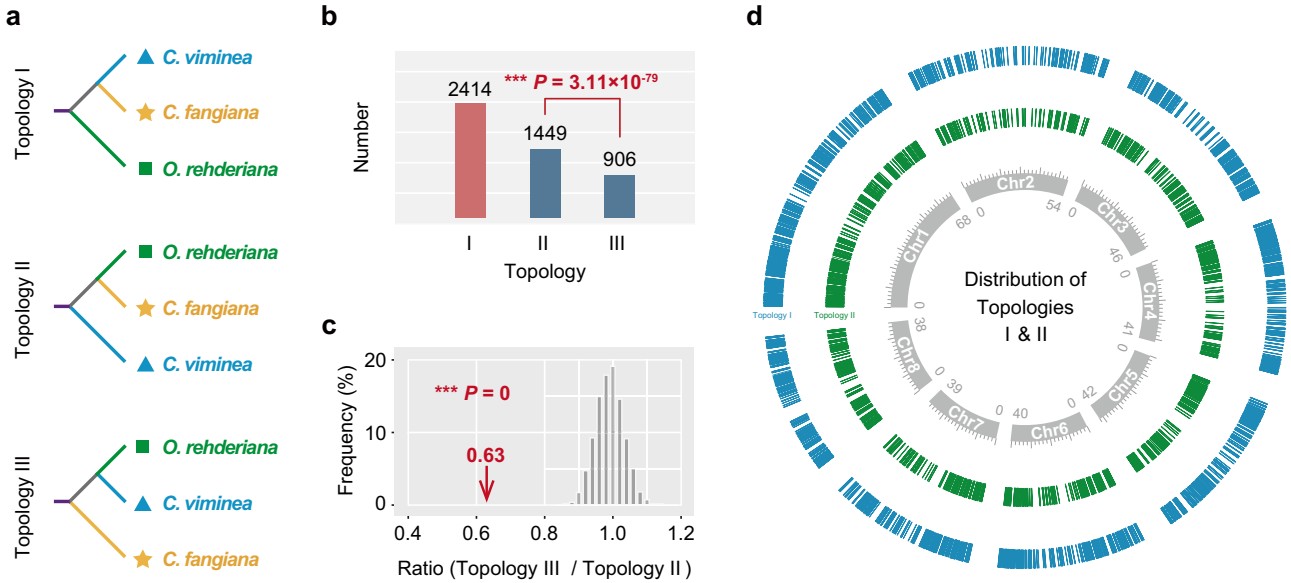

**Fig. 3 High incongruence in gene-tree phylogenies. a, b** Phylogenetic topologies (**a**) and the corresponding numbers (shown on the top of bar charts in (**b**) revealed from the phylogenies of 4769 strictly orthologous nuclear gene pairs with high-confidence support values (≥50). The number for Topology II was significantly more ($P = 3.11 \times 10^{-79}$) than that for Topology III. **c** Simulations under solely ILS scenario. The red arrow indicates the observed ratio (Topology III/Topology II) from 4769 ortholog groups (see in **b**). The gray bars are a histogram of the ratios obtained in 10,000 simulated datasets. The solely ILS hypothesis was strongly rejected due to a significant difference ($P = 0$) between observed and simulated ratios. **d** Genome-wide distribution of Topologies I (blue bars) and II (green bars). Statistical significance was determined by a two-tailed one-sample binomial test (**b**) and a two-tailed one-sample Student's $t$ test (**c**). Significant differences are indicated with asterisks (***$P < 0.001$). Source data are provided as a Source Data file.

(Supplementary Note 5). The maximum likelihood (ML) tree (Fig. 2c) of all re-sequenced individuals supported the monophyly of each lineage and agreed with the coalescent-based species tree (Fig. 2b). The times of divergence between the three lineages were dated from $K$s distributions for the representative species (*C. viminea*, *C. fangiana*, *O. rehderiana*, and *Betula pendula* as outgroup) based on a secondary calibration (Fig. 2d; Supplementary Note 4). The genera *Ostrya* and *Carpinus* were estimated to have diverged at 23–33 million years ago (mya) and the divergence time between sects. *Carpinus* and *Distegocarpus* was dated to 17–26 mya (Fig. 2c, d).

Then we explored the gene topologies of the three representative species, *C. viminea*, *C. fangiana*, and *O. rehderiana*. Phylogenies based on 4DTv sites from a total of 4769 ortholog groups with high-confidence support values (≥50) were obtained, comprised of three topologies (Fig. 3a, b; Supplementary Note 4). The most common tree (from 2414 ortholog groups) recovered *C. fangiana* as sister to *C. viminea* (Topology I). However, 1449 ortholog groups indicated that *C. fangiana* was sister to *O. rehderiana* (Topology II), significantly more ($P = 3.11 \times 10^{-79}$) than the number for topology III (906), in which *C. fangiana* clustered as a separate lineage. The unequal proportions of the three topologies suggest a likely hybrid origin for *C. fangiana*, since the latter two (Topologies II and III) would be expected to be nearly equal under a solely ILS scenario[11,29]. To further examine these significant differences, we simulated the gene trees and the proportion of each topology under the effects of ILS (Supplementary Note 4). The solely ILS hypothesis was strongly rejected due to a significant difference ($P = 0$) between observed and simulated ratios (Topology III/Topology II) (Fig. 3c). The genome-wide even distribution of Topologies I and II may further confirm the HHS hypothesis and exclude the solely introgression hypothesis (Fig. 3d and Supplementary Table 17).

**HHS test based on long indels from population genomic data.** Introgression usually leads to genetic mixture in a few individuals

or populations, rather than in all hybrid offspring as does HHS[2,5] (Supplementary Fig. 1 and Supplementary Data 1). We used population genomic data from the three lineages to verify the occurrence of an HHS event between the ancestors of the two assumed parental lineages before species diversification in each lineage. According to previous studies, indels (especially those ≥5 bp) were a type of homoplasy-free markers, which could minimize the interference from random and convergent nucleotide mutations during the long-term evolutionary histories of the three lineages[11–13]. Based on long indels (≥5 bp) from population genomic data, ABBA-BABA test ($D$-statistic)[30] and HyDe[31] were used to examine the likelihood of ancient hybridization following two previous methods[9,16] (Supplementary Note 6). These methods have been verified as being powerful for testing hybridization with the occurrence of ILS as a basis for shared polymorphisms[29,31,32]. When population genomic data are used, such methods can further distinguish HHS from introgression based on whether a hybridization signal is present in all hybrid offspring. The ABBA-BABA test indicated significant gene flow between sect. *Distegocarpus* and *Ostrya* ($D = 0.16$, $Z = 15.68$, $P = 0$; Fig. 4a) because of ancient hybridization rather than ILS only. HyDe also supported the hybrid origin of sect. *Distegocarpus* ($Z = 4.00$, $P = 3.16 \times 10^{-5}$), with a major genomic contribution (89%) from sect. *Carpinus* and a minor contribution (11%) from *Ostrya* (Fig. 4b). The hybridization event might have occurred around 17–33 mya, based on dating of the divergences of the representative species for ancestral lineages (Figs. 2c, d, 4b). We further extracted a total of 60,487 inter-group fixed long indels (≥5 bp) to exclude the ILS only and introgression hypotheses. Ancestral variations (AVs) and phylogenetically informative variations (PIVs) (reflecting the true evolutionary relationship) were detected between the members of each pair of selected lineages (Fig. 5a; Supplementary Note 6). If significant PIV signals could be detected in both the group consisting of sects. *Carpinus* and *Distegocarpus* (termed "CD") and the group comprising *Ostrya* and sect.

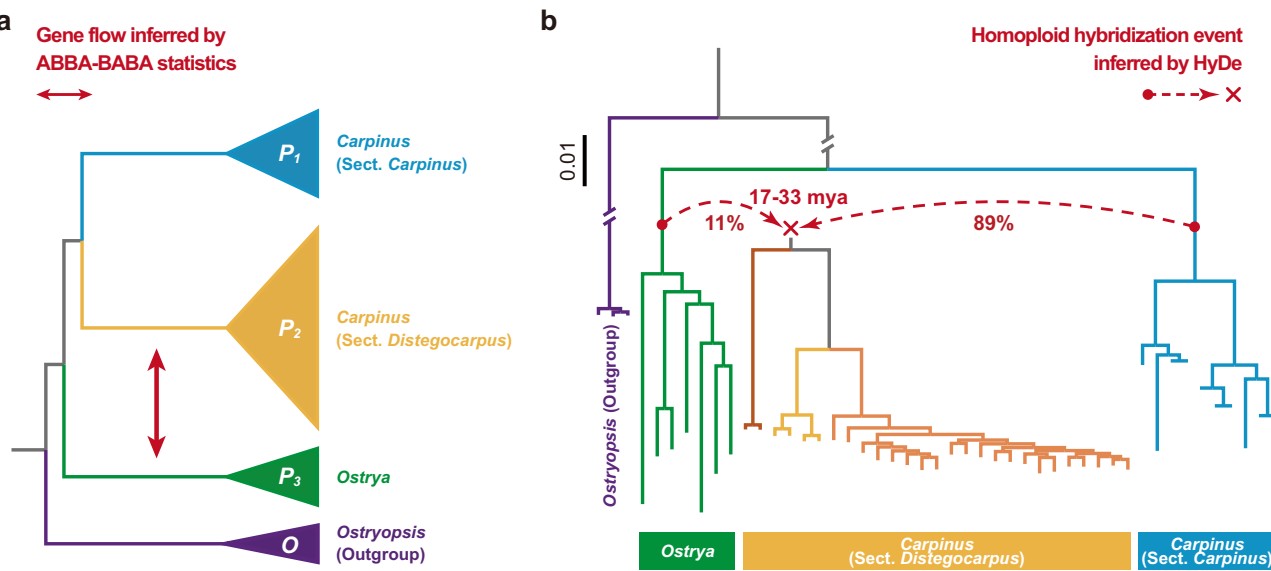

**Fig. 4 ABBA-BABA test and HyDe analysis using indels from population genomic data. a** ABBA-BABA test revealed the ancient gene flow between *Ostrya* and sect. *Distegocarpus*. Information for the lineages used was shown in the figure. **b** Homoploid hybrid origin of sect. *Distegocarpus* inferred by HyDe analysis. The red arrows (indicating the putative parents and hybrid lineages) and the red numbers (genomic contribution) indicate the results produced by HyDe. The time of homoploid hybrid origin was inferred by the previously dated divergence times between the three lineages (see in Fig. 2c, d). ABBA-BABA test and HyDe analysis were both performed at the population level, with the information from all individuals per population imputed together.

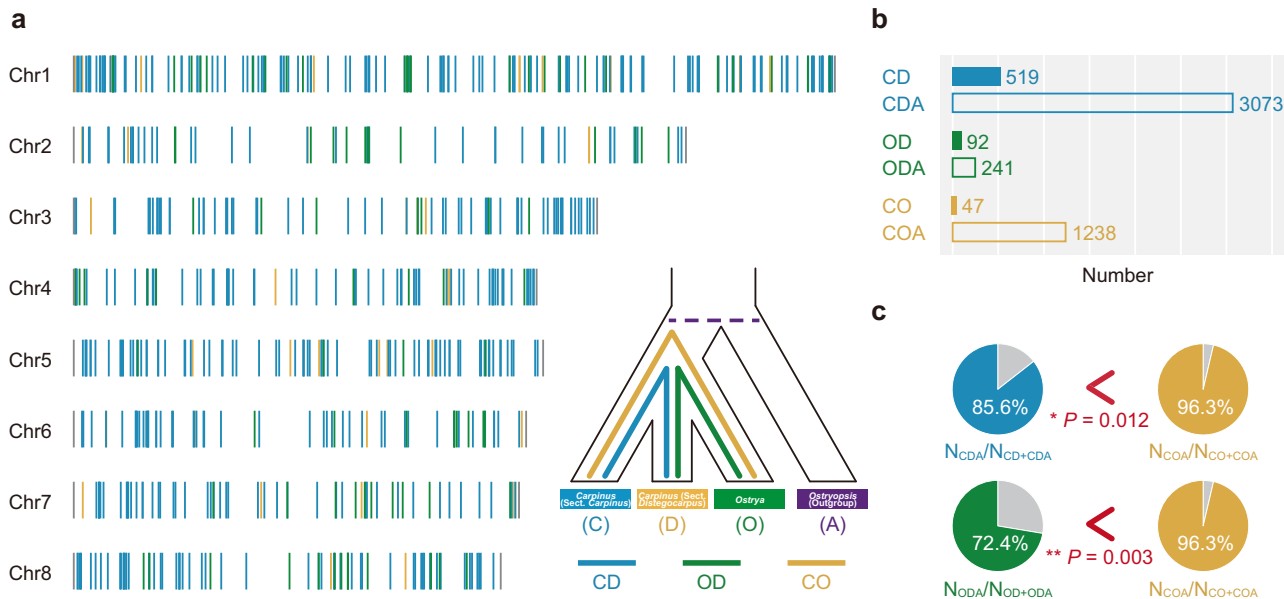

**Fig. 5 PIVs across the genome. a** PIVs identified across the genome based on population genomic data. **b** Number of PIVs ("CD", "OD", and "CO") and AVs ("CDA", "ODA", and "COA") shared by different combinations of lineages. The exact numbers are shown on the right of bar charts. **c** Significant PIV signals were detected in both groups (P = 0.012 and 0.003 respectively), revealing the homoploid hybrid origin of sect. *Distegocarpus*. Statistical significance was determined by a Pearson's Chi-square test with Yates' correction for continuity. Significant differences are indicated with asterisks (*P < 0.05; **P < 0.01). Sect. *Carpinus*, sect. *Distegocarpus*, *Ostrya*, and *Ostryopsis* (outgroup) are identified as "C", "D", "O", and "A", respectively. Different combinations of the letters indicated PIVs ("CD", "OD", and "CO") and AVs ("CDA", "ODA", and "COA") shared by different combinations of the lineages. The different colors denote sect. *Distegocarpus* sharing indels (PIVs or AVs) with sect. *Carpinus* ("CD" or "CDA", blue), *Ostrya* ("OD" or "ODA", green), or sect. *Carpinus* sharing indels with *Ostrya* ("CO" or "COA", yellow) as the graphic examples show. Source data are provided as a Source Data file.

*Distegocarpus* ("OD"), the hybridization hypothesis would be significantly favored. We found that PIVs shared by "CD" were the most abundant (519), followed by those shared by "OD" (92), in both cases being evenly distributed across the genome (Fig. 5a, b). The significant PIV signals for both "CD" and "OD" at the population level clearly support the hybrid origin of sect. *Distegocarpus* (Fig. 5c; Supplementary Note 6). In addition, these

abundant even-distributed long indels (≥5 bp) exclude the possibility of convergent homoplasy and introgression.

**Positively selected genes and hybrid signal.** Finally, we explored whether hybridization contributed to RI and intermediate morphological trait of nutlet bract of sect. *Distegocarpus*. It has been suggested that the likely hybridization-contributed RI can further

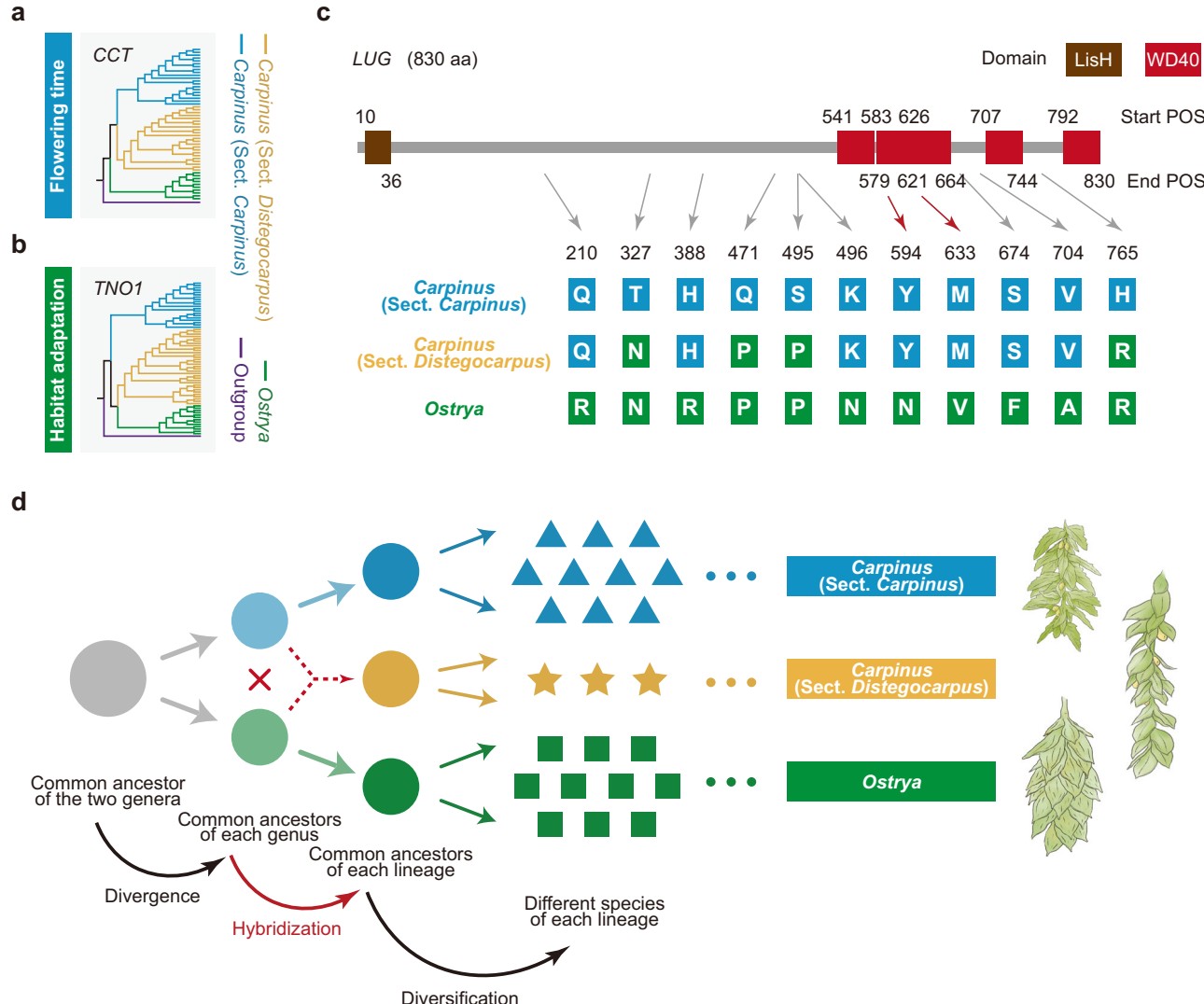

**Fig. 6 Selected PSGs involved in RI barriers and the *LUG* gene showing hybrid signals. a, b** Two selected RI-related PSGs of sect. *Distegocarpus* derived from sect. *Carpinus* (**a**) and *Ostrya* (**b**) respectively. ML trees were constructed based on haplotypes of protein sequences from all re-sequenced individuals. Each branch represents one identified haplotype. **c** The *LUG* gene showing hybrid signals because of hybrid recombination. All inter-lineage fixed amino acid differences were shown. **d** A simplified model of an ancient intergeneric HHS event between two genera and further species diversification. Representative morphologies of nutlet bracts from three lineages are shown on the right (Top: sect. *Carpinus*; middle: sect. *Distegocarpus*; bottom: *Ostrya*). In (**a**–**c**), the different background colors indicate the alleles (**a**, **b**) and mutations (**c**) of sect. *Distegocarpus* derived from sect. *Carpinus* (blue) and *Ostrya* (green).

distinguish HHS from introgression an/or general genetic admixture[3]. Genes controlling prezygotic RI between parental lineages experienced rapid evolution because of natural selection and the diverged alleles might be alternately fixed in the hybrid offspring lineage[5]. We therefore identified positively selected genes (PSGs) in the hybrid lineage (which were shared and derived from each parental lineage) following Wang et al.[5] (an integrated pipeline comprising HKA tests, non-synonymous mutations, and phylogenetic origin), which is the only pipeline hitherto developed to detect PSGs in HHS lineages and has been verified as effective by transgenic and common garden experiments (Supplementary Note 7). A total of 218 PSGs in sect. *Distegocarpus* were found to derive from sect. *Carpinus*, while 73 PSGs from *Ostrya* and functional analyses suggested that some of them are involved in flowering time and environmental adaptation, two crucial types of prezygotic RI in plant speciation process[4,33,34] (Supplementary Data 5, 6). For example, a flowering time-related *CCT* gene[35] in sect. *Distegocarpus* was derived from sect. *Carpinus* (Fig. 6a), while the *TNO1* gene, which is involved in habitat

adaptation[36], originated from *Ostrya* (Fig. 6b). In addition, we identified 19 genes in sect. *Distegocarpus*, each of which contained alternative amino acid mutations fixed by each of the two parental lineages (Supplementary Data 7). One of such genes, *LUG*, was reported to regulate floral organ development[37–39]. The *LUG* gene of sect. *Distegocarpus* inherited respectively 7 or 4 amino acid mutations which have been fixed in sect. *Carpinus* or *Ostrya* (Fig. 6c). This is partly consistent with the intermediate nutlet bract of sect. *Distegocarpus*, because nutlets of *Ostrya* are completely enclosed by bracts, while those of sect. *Carpinus* are rarely enclosed by bracts[19] (Fig. 6d). This recombinant *LUG* gene and others that experienced similar recombination due to hybridization may together have led to the intermediate nutlet bracts in sect. *Distegocarpus*.

## Discussion

In this paper, we report a de novo genome sequence for a *Carpinus* species and 44 re-sequenced genomes for representative

individuals from two sections of this genus and *Ostrya*. Our analyses of multiple genomic datasets revealed highly discordant gene topologies (Fig. 3 and Supplementary Table 17) between sect. *Distegocarpus* and sect. *Carpinus* and *Ostrya*, which may have resulted from homoplasy, ILS, introgression, or HHS. We found that numerous long indels across the whole genome showed such inconsistent relationships and were shared by sect. *Distegocarpus* with sect. *Carpinus* or *Ostrya*, a result that seems unlikely to be explained on the basis of evolutionary homoplasy[11,12]. We excluded ILS as an explanation of observed patterns based on an ABBA-BABA test (*D*-statistic)[30] and HyDe analysis[31] (Fig. 4), both of which have been proved to be powerful methods for detecting hybridization[29,31,32]. Under both ancient and recent introgression scenarios, only a few hybrid offspring in sect. *Distegocarpus* would be expected to contain genomic admixture from the presumed parents[9] (Supplementary Data 1). However, we found that long indels (PIVs) specific to sect. *Carpinus* and to sect. *Ostrya* were present together and fixed in all sampled species and individuals of sect. *Distegocarpus* (Fig. 5a). The hybridization that led to a genomic admixture in all samples of sect. *Distegocarpus* is therefore likely to have occurred earlier than the further diversification of this section into three current species. In addition, the major distinction between introgression and HHS is whether a stable and distinct lineage had been established[9]. These findings seem to suggest that sect. *Distegocarpus* has not only evolved as an independent and stable lineage but also diversified into three present-day species.

In addition to genetic admixture, we found that the ancient hybridization appears to have directly created RI in sect. *Distegocarpus*, initiating its independent evolution, based on the following findings. First, we found that some genes responsible for critical prezygotic RI (e.g., flowering time and habitat adaptation) were derived from either one or the other of the two parental lineages (Fig. 6a, b and Supplementary Data 5, 6), especially those genes that had experienced strong selection pressure and showed signals of positive evolution. The hybrid recombination of these RI alleles probably could have rapidly developed prezygotic RI in hybrids at the initial HHS stage by giving rise to differences in flowering time and habitat adaptation[5], although further functional testing of these alleles to RI is needed in the future. Second, the alternatively fixed long indels (PIVs in Fig. 5a) from the two parents in numerous loci across the whole genome could have also created immediate postzygotic RI based on Bateson-Dobzhansky-Muller (BDM) genetic incompatibility[7]. With these initial RIs created by hybridization, the overall RI between sect. *Distegocarpus* and each parental lineage could have accumulated continuously and been reinforced in the course of its independent evolution. Thus the three strict criteria for the HHS hypothesis appear to be met[3]. In addition, recombinant genes formed during HHS may have accounted for some of the intermediate morphological traits of sect. *Distegocarpus* (Fig. 6c, d). The origin of sect. *Distegocarpus* was dated at around 17–26 mya, only a few million years after the divergence of the two parental lineages about 23–33 mya. Sect. *Distegocarpus* therefore likely originated through HHS during the early divergence between *Carpinus* and *Ostrya*.

Phylogenetic discordance based on different genetic datasets has been widely reported for plants at higher taxonomic levels, from genus to family, order and angiosperm clades[40–43]. Because WGD has been found to occur in numerous specific lineages (e.g., core eudicot), allopolyploid hybrid speciation has sometimes been invoked to explain such discordant phylogenies[44–46]. However, our results suggest that HHS without WGD during the initial divergence of these higher taxonomic lineages may also lead to phylogenetic discordance. In the future, this alternative explanation could be considered and tested based on the increasingly available genomic resources.

## Methods

**Genome sequencing and assembly of *C. viminea*.** Fresh leaves of a wild *C. viminea* individual were collected from Ya'an, Sichuan Province, China (102°45′E, 30°23′N). Total genomic DNA was extracted using the cetyltrimethyl ammonium bromide (CTAB) method[47]. For sequencing, the Nanopore sequencing library, paired-end Illumina library, and Hi-C library were constructed and then sequenced by a PromethION DNA sequencer (Oxford Nanopore, Oxford, UK), a MGISEQ-2000 platform (Illumina, San Diego, CA, USA), and an Illumina HiSeq 4000 platform (Illumina, San Diego, CA, USA), respectively. We also collected four fresh tissue samples (leaf, flower, bud, and twig) from the same *C. viminea* individual for total RNA sequencing. The Nanopore long reads were corrected using NextDenovo (https://github.com/Nextomics/NextDenovo) and de novo assembled using SmartDenovo (https://github.com/ruanjue/smartdenovo). The contigs were corrected and polished with the Illumina reads using Pilon[48] for three rounds. HiC-Pro[49] was then used to analyzed and assessed the Hi-C data. LACHESIS[50] was employed to cluster, reorder, and orientate the corrected contigs into pseudochromosomes.

**Genome annotation of *C. viminea*.** RepeatMasker v.4.0.7[51] and RepeatModeler v.1.0.10 (http://www.repeatmasker.org) were used to annotate the repeat sequences. For gene prediction, PASA v.2.1.0 (Program to Assemble Spliced Alignments)[52], Augustus[53], and GeneWise v.2.4.1[54] were performed, respectively. EVidenceModeler (EVM) v.1.1.1[55] was then used to combine these results, which generated the final protein-coding gene set. For functional annotation, the predicted genes were searched against Swiss-Prot[56], TrEMBL[56], and NCBI non-redundant protein (NR)[57] databases. InterProScan v.5.25-64.0[58], blast2GO v.4.1[59], and KEGG (Kyoto Encyclopedia of Genes and Genomes) Automatic Annotation Server (KAAS)[60] were also performed, respectively.

**Genome improvement of *O. rehderiana*.** To further extend the contiguity of the *O. rehderiana* genome assembly[25], we collected a fresh sample of *O. rehderiana* from Tianmu Mountain, Zhejiang Province, China (119°27′E, 30°20′N) for Hi-C sequencing and performed chromosome anchoring following the same pipeline as those for the *C. viminea* genome assembly.

**Genome features and comparative genomic analyses.** For *C. viminea* assembly, genome features were summarized by a 500 Kb non-overlapping sliding-window and then visualized by Circos[61]. We performed the comparative genomic analyses with the representative species of different Betulaceae genera, including *C. viminea*, *C. fangiana*[26], *O. rehderiana*, *Ostryopsis davidiana*[5], *Corylus mandshurica*[27], and *Betula pendula*[28]. *Casuarina equisetifolia*[62] and *Juglans regia*[63] were used as the outgroups. OrthoFinder[64] and PRANK[65] were used to identify the strictly orthologous nuclear gene groups between the eight species (1:1:1:1:1:1:1:1) and aligned their coding sequences (CDS) respectively. RAxML[66] was then used to construct the maximum likelihood (ML) tree using a concatenated matrix of 4DTv (fourfold synonymous transversion) sites. MCScanX[67] was employed to identify the synteny blocks between the six Betulaceae species (involving ≥20 collinear genes) and within each of them (involving ≥5 collinear genes). LAST[68] was also used to perform the collinearity analysis between *C. viminea*, *C. fangiana*, and *O. rehderiana* based on their whole-genome sequences.

**Hybridization test based on de novo genome sequences.** We performed the hybridization tests between the three species, *C. viminea*, *C. fangiana*[26], and *O. rehderiana*, with *Ostryopsis davidiana*[5] as outgroup. OrthoFinder[64] was employed to identify the strictly orthologous nuclear gene groups (1:1:1:1) based on the protein-coding sequences. For each group of genes, PRANK[65] was used to align their coding sequences (CDS). Then, we extracted their 4DTv sites, neutral sites of a widely used type which are capable of minimizing interference due to balance selection. RAxML[66] was employed to construct the ML tree for each gene group based on their concatenated 4DTv sites. ASTRAL[69] was used to estimate the species tree under a multi-species coalescent model. The branch lengths of the species tree so generated were in coalescent units. According to the coalescent-based species tree produced, DendroPy[70] was then applied to simulate the gene trees under the effects of ILS. Finally, to infer the time scale of the hybridization event, we estimated the times of divergence based on the distributions of *Ks* (synonymous substitutions per synonymous site) values and a secondary calibration.

**Population materials.** To explore more genetic detail, we sampled 47 individuals (including a total of 21 species) from all of three lineages for population genomic resequencing, including ten individuals (from ten species) of sect. *Carpinus*, 27 individuals (from three species) of sect. *Distegocarpus*, seven individuals (from seven species) of *Ostrya*, and three individuals (from one species) of *Ostryopsis* as the outgroup. Except for the outgroup, all sampled individuals were selected from different populations (one individual per population). The samples of sect. *Carpinus* and *Ostrya* covered all acknowledged species of these two lineages, respectively. The samples of sect. *Distegocarpus* covered all three species described for this lineage and almost all samples were collected in natural field.

**Genome resequencing, read mapping, and variants calling**. For each sample, total genomic DNA was extracted by the CTAB method[47]. Illumina paired-end reads were produced by Illumina HiSeq platform (Illumina, San Diego, CA, USA). The filtered reads were then mapped to the reference genome, the genome of *C. viminea*, by BWA-MEM[71] with recommended parameters. SAMtools[71] was used to sort the mapped reads and further remove the duplicated ones. The generated BAM files were used for variant calling based on the pipeline corresponding to GATK best practice. GATK HaplotypeCaller[72] was first performed to detect the variants for each sample. Then, the variants identified for each sample were merged by GATK GenotypeGVCFs[72]. After a set of robust filtering, we obtained a high-quality SNP set, containing 6,244,030 biallelic SNPs with the outgroup and 6,302,136 biallelic SNPs without the outgroup. We also identified a total of 443,792 long indels (≥5 bp) (with the outgroup) for subsequent analyses.

**Hybridization test based on population genomic data**. To explore the hybridization event in greater depth, we employed hybridization tests using the population genomic data. The population-level phylogeny was constructed using RAxML[66] using the previously generated 6,244,030 biallelic SNPs. The other analyses were performed based on the previously identified 443,792 long indels (≥5 bp). First, HyDe[31] was performed to test whether a hybrid origin scenario was supported. The individuals were classified into four groups: sect. *Carpinus*, sect. *Distegocarpus*, *Ostrya*, and *Ostryopsis* (as the outgroup). Then, Dsuit[73] was used to perform the ABBA-BABA test (*D*-statistic)[30], with sect. *Carpinus*, sect. *Distegocarpus*, *Ostrya*, and *Ostryopsis* as $P_1$, $P_2$, $P_3$, and $O$, respectively. HyDe and ABBA-BABA tests were performed at the population level, with the information from all individuals per population being input together. Finally, to test the ancient HHS scenario, we modified the approach developed by Jiang et al.[11] and applied it to the population-level data (Supplementary Note 6). We identified the AVs and the PIVs across the genome and further detected the significant PIV signals. AVs and PIVs were classified based on their times of occurrence[11]. AVs occurred before the differentiation of all species. PIVs occurred after the first species differentiated and before the last one. If significant PIV signals can be detected between the assumed hybrid species and each of the parental species, the HHS assumption is validated and introgression and ILS can be excluded.

**Positively selected genes and hybrid signals detected on genes**. We identified PSGs and genes harboring hybrid signals because of hybrid recombination based on the re-sequenced individuals (using the previously identified 6,302,136 biallelic SNPs without an outgroup), respectively. PSGs were identified following the methods in Wang et al.[5] (Supplementary Note 7), where all final PSGs must conformed to three criteria: (1) significant *P* values (≤0.01) in HKA tests[74]; (2) the number of fixed non-synonymous mutation sites ranked in the top 2.5% of all genes tested; (3) phylogenies with sect. *Distegocarpus* individuals deriving mainly from one parental lineage. The hybrid signals were detected according to whether the genes were positively selected with both grouping methods: sects. *Distegocarpus* (hybrid lineage) and *Carpinus* as one group compared with *Ostrya*, and sect. *Distegocarpus* and *Ostrya* as one group compared with sect. *Carpinus*. When detecting hybrid signals, only two criteria (a significant *P* value in an HKA test and the number of fixed non-synonymous mutation sites ranking in the top 2.5%) were considered, taking no account of phylogenies.

**Reporting summary**. Further information on research design is available in the Nature Research Reporting Summary linked to this article.

## Data availability

All the sequencing data have been deposited in the National Genomics Data Center (https://ngdc.cncb.ac.cn). The sequencing data for the de novo genome of *Carpinus viminea* (including the ONT long reads, Illumina reads of WGS, Hi-C reads, and transcriptomes) have been deposited under the accession number PRJCA005724. The Hi-C sequencing data for *Ostrya rehderiana* have been deposited under the accession number PRJCA005717. The resequencing data for all individuals have been deposited under the accession numbers PRJCA003130 and PRJCA005842. The genome assemblies and annotations of *C. viminea* and *O. rehderiana* have been uploaded to figshare [https://doi.org/10.6084/m9.figshare.14988777]. Source data are provided with this paper.

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

## Acknowledgements

This work was supported equally by the Strategic Priority Research Program of the Chinese Academy of Sciences (XDB31000000), National Natural Science Foundation of China (grant numbers 31590821 and 91731301) and the National Key Research and Development Program of China (2017YFC0505203), and also by Fundamental Research Funds for the Central Universities (SCU2019D013 and 2020SCUNL207), and National High-Level Talents Special Support Plan (10 Thousand of People Plan).

## Author contributions

J.Liu designed and led the project. Z.W. conducted the research. C.C. collected the materials. Z.W. prepared DNA for sequencing and analyzed the data. M.K. contributed to genome assembling and prediction. J. Li, Z.Z., Y.W., and Y.Y. provided technical support. Z.W. wrote the manuscript. J.Liu revised the paper. Z.W. and M.K. contributed equally.

## Competing interests

The authors declare no competing interests.
