## [Peer Review File · Nature Communications]

Genomic evidence for homoploid hybrid speciation between ancestors of two different generaReviewers' Comments:

Reviewer #1:

Remarks to the Author:

There is much of interest in this MS, including a better phylogenetic understanding of the relationships of *Ostrya* and *Carpinus*, evidence for rampant gene tree incongruence, and a large body of new genome sequence data for the *Betulaceae*. However, I am not convinced that the work supports the conclusion stated by the title of the paper. I think that incomplete lineage sorting, homoploid hybrid speciation and ancient introgression all remain as possible explanations for the gene tree incongruence that the authors have found.

Clearer and more detailed introduction is needed on the taxonomy and existing phylogenetic understanding of the taxa being studied, and especially the section *Distegocarpus*. Is this the section *Distegocarpus* of Winkler (1904) or Li and Cheng (1979)? An authority is needed on line 32. The three species that make up this section need to be named early in the paper. The statement "sect. *Distegocarpus* ... exhibits intermediate morphology between these two genera and has an unsolved systematic position [10–13]" is inadequate. Reference 12 (Ma et al 2015) seems to be mis-cited as it does not include any species from *Distegocarpus*. Reference 11 (Yoo & Wen 2002) shows *Carpinus cordata* and *C. fangiana* (section *Distegocarpus*) as sister to a clade with the rest of *Carpinus* and *Ostrya* in an ITS analysis, but with *Carpinus japonica* (section *Distegocarpus*) as sister to them all. Reference 13 (Yang et al 2019) Shows *Ostrya* as sister to *Carpinus* in plastome tree but nested within *Carpinus* in an ITS tree. The authors need to cite Li JH. 2008 [Sequences of low-copy nuclear gene support the monophyly of *Ostrya* and paraphyly of *Carpinus* (*Betulaceae*). *Journal of Systematics and Evolution* 46: 333–340] whose analyses show *C. cordata* as sister to the rest of *Carpinus* plus *Ostrya* (with *C. japonica* grouping with the other *Carpinus* species.): thus, in this uncited study, Section *Distegocarpus* is not monophyletic. These existing phylogenetic treatments seem to suggest that *Ostrya* has arisen from within *Carpinus*. There does not seem to be an a priori reason to suspect from the existing analyses that Section *Distegocarpus* arose via hybridisation between *Ostrya* and *Carpinus* sect. *Carpinus*. It is therefore unclear to me why the authors set up their study as a test of the hypothesis that *Distegocarpus* had a hybrid origin. It seems to me that this was a conclusion that arose from their phylogenomic study, rather than its starting hypothesis. To my mind the MS would be much clearer if it was written in this way: i.e. as a phylogenomic study on the unresolved relationships of *Carpinus* and *Ostrya*.

From a phylogenomic analysis, the authors provide very good evidence that the majority of genes support the monophyly of *Carpinus* sect. *Disegocarpus* and *Carpinus* sect. *Carpinus*, with *Ostrya* as a sister. They also give evidence for extremely high gene-tree incongruence among *Ostrya*, *Carpinus* sect. *Disegocarpus* and *Carpinus* sect. *Carpinus*. (Figure 1 b and c). These in themselves are interesting results and deserve to be highlighted more explicitly.

The authors then seek to understand if this extreme gene-tree incongruence is due to incomplete lineage sorting (ILS) or introgressive hybridisation or homoploid hybrid speciation or convergent evolution (homoplasy). They claim to show that only homoploid hybrid speciation can explain the patterns they find.

To eliminate ILS, the authors assume that under ILS, similar numbers of genes would show the topologies II and III of Figure 1b. I can see how this would be true if all the genes were evolving in a neutral fashion, or we were looking at a short timespan since the lineages split. However, we are looking at a timespan estimated to be around 30 million years, and the authors later present evidence for the action of biased selection (lines 91-92). I am not therefore convinced that they can eliminate ILS as a hypothesis.

To eliminate introgressive hybridisation the authors analyse genome sequences for many different species or individuals within *Ostrya*, *Carpinus* sect. *Disegocarpus* and *Carpinus* sect. *Carpinus*. They

argue that if introgression had occurred since speciation occurred within these three groups, then different species within the three groups would show different levels of introgression. This argument is valid, but it does not eliminate the hypothesis of introgressive hybridisation occurring before the ancestor of the three species within *Carpinus* sect. *Disegocarpus* began to speciate. Even if all three species of *Carpinus* sect. *Disegocarpus* show similar levels of introgression from *Ostrya*, this could be due to introgression into their common ancestor. In other words, the ancestors of *Carpinus* sect. *Disegocarpus* and sect. *Carpinus* could have speciated before introgression from *Ostrya* into the ancestors of sect. *Disegocarpus* occurred. Thus, homoploid hybrid speciation between the ancestors of *Ostrya* and *Carpinus* sect. *Carpinus* is indistinguishable from introgressive hybridisation between *Ostrya* and an already diverged ancestor of *Carpinus* sect. *Disegocarpus*.

In any case, I do not find Figure 1f (see also line 68) clear evidence against introgression since the divergence of sect. *Disegocarpus*, as even if some recent introgression had occurred, this topology could have been found, based on genome-wide SNPS (line 177). The ABBA-BABA test, HyDe and HHS tests seem to provide more compelling evidence that hybridisation occurred prior to the divergence of the species within section *Disegocarpus* (though the methods here need to be explained more clearly). But they do not (as far as I can see) distinguish between homoploid hybrid speciation and introgressive hybridisation between *Ostrya* and an already diverged ancestor of *Carpinus* sect. *Disegocarpus*.

The hypothesis of convergent evolution is given less attention by the authors. They seek to deal with this using long indels (line 72) as markers in the hybridisation tests. They could say more about why they expect these to be unlikely to have evolved convergently. But overall, I agree with them that convergent evolution is very unlikely to underlie the gene tree incongruence that they find.

The final section of the MS (lines 86 to 114) is built on the assumption that the authors have conclusively demonstrated that homoploid hybrid speciation has occurred. As I am unconvinced that they have demonstrated this, this section does not convince me either. Some of the patterns they report on genes under selection are nonetheless interesting – I would just like to see them discussed in a framework of gene tree incongruence, rather than homoploid hybrid speciation.

There are numerous linguistic infelicities throughout the paper that also need to be worked on.

The authors may also wish to consult these two papers:

Guo, Xing, Daniel C. Thomas, and Richard MK Saunders. "Gene tree discordance and coalescent methods support ancient intergeneric hybridisation between *Dasymaschalon* and *Friesodielsia* (Annonaceae)." *Molecular Phylogenetics and Evolution* 127 (2018): 14-29. This seems to be a somewhat similar paper to the present MS.

Crane, P. R. "Betulaceous leaves and fruits from the British Upper Palaeocene." *Botanical Journal of the Linnean Society* 83.2 (1981): 103-136. This reports a fossil that shows a "combination of the characters of several living genera, particularly *Corylus* and *Carpinus*"

Reviewer #2:

Remarks to the Author:

This paper analyses genome level data from multiple tree species in two genera and identify a clade that has originated through hybridization between genera. While studies confirming homoploid hybrid speciation are becoming more frequent thanks to the resolution provided by genomic data, the novelty of this study is to detect an older hybridization event predating generic divergence. This is an advance of interest.

The approach using orthologous gene sets and indels seem robust to detect hybridization and

distinguish it from incomplete lineage sorting and gene flow. The new reference *Caprinus* genome is built from extensive sequence data and is likely to be highly reliable. The inclusion of an improved *Ostrya* genome is valuable to prevent ascertainment biases influencing subsequent analyses.

However, later in the methods, I'm disappointed to learn that population samples were mapped against just one of the available genomes, missing an opportunity to protect against ascertainment bias. The authors should justify this approach.

The hypothesis to distinguish HHS from introgression needs to be better explained as it is vital for the reader to assess the subsequent results and interpretations. Also tests such as HyDe are presented to support HHS over introgression without sufficient information for the reader about how this distinction can be made.

The discussion about selection in the new hybrid lineage is interesting but the results as presented have been over-interpreted. I understand the evidence to be that some genes are alternately fixed in the hybrid lineage. This is not sufficient evidence in itself for either selection or reproductive incompatibilities.

The language used is awkward but mostly understandable so it would be worth having the paper edited by a native language speaker.

Overall, this is an exciting study system to demonstrate an older HHS event but the paper needs to present stronger evidence of HHS versus introgression to be really convincing. This could simply be a case of carefully explaining the tested hypotheses and the logic of the tests applied. Not all researchers in the field argue for such a strong distinction to be made between introgression and HHS, but simply demonstrating past introgression as part of diversification would be less novel in my opinion.

Specific comments

L11-13 Indels incorrectly capitalized here and elsewhere. I would replace "of" with "from" here. I'm not sure how indels data can exclude hypotheses of introgression.

L32 State the genus that sect. *Distegocarpus* is currently assigned to, or if there is controversy. It is important to clearly explain the background to the system to help readers evaluate the advances of the study.

L40 It would be worth stating the Hi-C sequencing coverage of the *O. rehderiana* genome here for comparison with the *C. viminea* genome.

L55-56 Mention and justify mutation rate and generation time used to estimate divergence times.

L56-58 Why is no divergence time between *C. fangiana* and *O. rehderiana* provided?

Figure 1 What species is represented by Bpe in panel e and why is it important to show it in this panel? The HyDe summary is too vague. State what is being quantified here.

L59-60 Why would not all hybrid offspring show signals of admixture? This hypothesis needs to be really clear.

L66 Which reference was used and why?

L71 I think you should briefly mention the lineages used for each of the ABBA-BABA and HyDe tests here.

L72 The phrase "at utmost" is not clear. I think you are arguing that indels are effective at excluding the hypothesis of homoplasy.

L74-75 Briefly explain how HyDe can distinguish HHS from introgression.

L87-89 Not all alleles will alternately fix due to selection on reproductive incompatibilities. Drift could play a role, as could directional selection on new hybrids.

L90 Similar to my previous point, the term positively selected genes would only be valid if there is a specific test of selection over and above being fixed in the new hybrid lineage. State the test applied or change this term to alternately fixed genes.

L99 I don't follow the distinction between mutations and alleles. From the following example, I think that you are referring to a combination of alternate mutations from each parental lineage within the same gene. Clarify the explanation here.

L105 Gene now named "LUC" rather than "LUG", one of these is a typo.

L135 Typo, change "predicated" to "predicted".

L161 Typo, change "almost samples" to "almost all samples"

L165-167 What was the reason for mapping all samples to *C. viminea*, when there was also the choice of the *O. rehderiana* genome? Can you justify that this would not lead to ascertainment bias in SNP calling?

L182-186 I recommend adding a little more detail on the logic of this relatively new test here about how it can distinguish between introgression and HHS.

L195-197 The last part of this methods needs to be rewritten for clarity.

Figure 2 The consistent lineage colours from Figure 1 are useful but the changed naming system is confusing. I prefer the C,D,O,A naming system of figure 2. Indicate which genome is shown in panel a, and which rows are PIVs and AVs in panel b. Panel c seems to show the crucial test of HHS. Explain this test in the main text.

Reviewer #3:

Remarks to the Author:

This paper looked at the evolution of a plant family to argue it is a product of homoploid hybrid speciation (HHS). I disagree with the authors claims and suggest a simple scenario where there's been hybridization between sect *Distegocarpus* (which the authors claim are the hybrid) and *Ostrya* (the third outgroup), without invoking any evidence of speciation is sufficient to explain all their results. And based on this result one does not have to invoke HHS at all for their study system. I'd suggest the authors to read the review from Schumer et al.

(<https://onlinelibrary.wiley.com/doi/10.1111/evo.12399>) for details on what HHS is, since simply having evidence of hybridization is not sufficient to invoke hybrid speciation.

Now I agree that the authors have evidence of a hybridization. Although I think some of the results are basically redundant to each other, for instance Fig 1C and Fig 2A,B is basically saying among three possible topologies for a rooted three taxon group, between the two minor topologies one is significantly more abundant than the other. I agree this is evidence of hybridization but not sure it warrants a separate figure for the results. But importantly still how can the authors eliminate the possibility that this is simply a hybridization from secondary contact? For this reason I think the evidence that this system is HHS is very weak.

Also they conduct evidence of positive selection, but the method they use (an HKA test) is very underwhelming. There are so many powerful tests that look for evidence of a selective sweep genome-wide and the lack of using any of these state-of-the art methods is a concern. In addition, anybody really can make a story out of false positive selective sweep results (see <https://academic.oup.com/mbe/article/29/10/3237/1032149>) that their positive selection results don't seem that convincing.

In sum, I think it requires more evidence than presented in this manuscript to show a certain lineage is a product of HHS. One that includes showing reproductive isolation and evolutionary genetic analysis that shows hybridization of the parental taxa, and not evidence of a hybrid swarm or secondary contact. But even if I give all the benefit of doubt for the authors, I think in the end the bottomline of this manuscript simply is that there is a lineage and it's a product of HHS, which is a pretty limited scope in my view.

I'd also suggest the authors could give biological background on their study system. The lack of any background also makes it hard as a reader to understand why this system is a potential case of HHS, and understand (or even sympathize) the authors case to present this system as a case of HHS. In fact I didn't even understand if this was an animal or plant system until I saw Fig3 plant cartoons.

Reviewer #4:

Remarks to the Author:

This manuscript describes the collection and curation of data that are then used to examine the question of ancient hybridization between distinct plant genera leading to formation of a new species. A variety of methods are applied to the genomic data: a simulation-based test based on tree topologies; the ABBA-BABA test; HyDe; and a more recently proposed test based on phylogenetic informative variations. The data sets developed and methods applied are interesting, and the question is one that is commonly asked with such data in plants. In this regard, the manuscript would appeal to a broad audience.

However, the manuscript at present is poorly written and difficult to read. In addition to many grammatical errors, there is not enough information given with regard to the methods used to test for hybridization. There's the potential these have been applied correctly, but it would be impossible for the reader to know and, most importantly, impossible for a reader to replicate and apply to their own data. I provide some specific examples below. Minor typos and grammatical errors are listed below that for the main text; other sections contain similar errors, but I didn't list them all.

Major issues:

1 — I don't understand why a simulation approach is being used (line 52). These are three-tip trees, right? The coalescent model specifies the probabilities of the three topologies in the absence of hybridization: the gene tree matching the species tree has probability $1 - 2/3 \exp(-t)$ and the other two topologies have probabilities $1/3 \exp(-t)$, where t is the length of the single internal branch in coalescent units. Why simulate when these probabilities could just be calculated? And, whether simulation or exact calculation is to be used, how is t chosen? This needs to be specified in the manuscript, as it will significantly impact the results.

2 — I think the authors spend too much time discussing homoplasy, given that the methods they use (e.g., HyDe) can handle homoplasy. Model-based methods like HyDe include the possibility of homoplasy as part of the substitution model. As an example, I don't understand the sentence (lines 28-29) "The well-assembled genomes and the accordingly extracted long [I]ndels can exclude most of such evolutionary homoplasy." How do they "exclude" it? Also, why is an indel of more than 5bp considered long? This seems quite short in terms of phylogenetic information.

3 — Much more detail needs to be provided about the implementation of the ABBA-BABA test and HyDe. First, it is unclear in Section 5.1 what the role of the estimated phylogeny is. Neither HyDe nor the ABBA-BABA test require this. What were the bootstrap support values used for? Why were the indels transformed into "pseudo-alleles"? HyDe handles full sequence data so this seems like a potential loss of information. What does "the analysis was performed at the population level" mean? Perhaps giving the specific HyDe commands used would help to clarify all of this. Similar comments apply for the ABBA-BABA test. Does the Dsuite software have options available? Which were used?

4 — The authors need to explain what they consider high support, and how support was measured.

Minor typos/grammatical errors:

- line 4: "... has been increasingly recognized widely occurred"
- line 6: change to: "focused on closely-related species"
- line 6: "while it" — what does it refer to? This modifies "previous studies" so "it" doesn't make sense

- line 11 and throughout: why is "Indels" capitalized?
- line 13: "at utmost" — what does this mean? Please re-word (this occurs elsewhere in the manuscript, as well)
- line 22 and line 24: "is" vs. "was" — watch tense
- lines 22-23: change to "to distinguish"
- line 26: re-word "difficulty to exclude homoplasy"
- line 27, remove "the"
- line 34, change "the" to "a"
- line 43, change to "The genomes of these three species"
- line 46, "were revealed" — re-word
- line 47, "high-confidence supports" — this is unclear
- lines 47 and 50: "the 2,414 ones" and "906 ones" — re-word
- line 51, remove "the"
- line 52: "the ILS hypothesis" — what does this mean? ILS is an evolutionary process. Please re-word. I think the authors mean the hypothesis that gene tree incongruence is due to solely to the process of ILS and not to hybridization.
- line 70, the phrase, "Based on the [I]ndel spectrum" is not clear.
- line 71: change to "used the ABBA-BABA test"
- line 73, remove the words "following" and "the"
- lines 80 and 81: "the most number" and "the second" — re-word
- line 87, what "intermediate morphological trait" is being referred to here?
- line 100, remove "of"
- line 107, change to "reveal a case of ancient intergeneric HHS"
- line 178 (and elsewhere), the wording "to detect the likely hybridization event" is odd. The authors are actually testing to see whether the data support hybridization, right?
- line 192, the phrase "the top 2.5% number of fixed non-synonymous mutation sites" is unclear
- line 196, "in both grouping ways" — this is unclear
- The title of Section 3.4 in the Supplementary Information is misleading. The authors have written, "The exclusion of the incomplete lineage sorting effects by simulation". But I think what they mean is that ILS *alone* cannot explain the patterns observed in the gene tree topologies.

Reviewers' comments:

Reviewer #1 (Remarks to the Author):

There is much of interest in this MS, including a better phylogenetic understanding of the relationships of *Ostrya* and *Carpinus*, evidence for rampant gene tree incongruence, and a large body of new genome sequence data for the Betulaceae. However, I am not convinced that the work supports the conclusion stated by the title of the paper. I think that incomplete lineage sorting, homoploid hybrid speciation and ancient introgression all remain as possible explanations for the gene tree incongruence that the authors have found.

Response: We really appreciate your constructive comments and suggestions. We are so sorry that our manuscript could not convince you and made you confused. Actually, the primary version of the manuscript was in the traditional 'Short Report' format, which may be too short to provide enough background of research field and methodological details (most of them were in the Supplementary Information). Here, we revised the manuscript according to your suggestions and expanded the revised manuscript to the full-length article format to demonstrate more details.

We have made great efforts on the major question - how to eliminate introgression (including ancient introgression) and the effects of incomplete lineage sorting (ILS). The more detailed background of homoploid hybrid speciation (HHS) was introduced, especially for the distinction between HHS and introgression (including ancient introgression) and the three key criteria for HHS. We carefully described our analyzing approaches, including how they can eliminate ILS effects and ancient introgression. We also fully discussed the rationality of our methods and the accuracy of our conclusions. To better illustrate the distinction between HHS and introgression, we added a new supplementary figure (Fig. S1) and provided a list of ~ 150 papers (Table S1) supporting our claim. We hope our efforts can let you convinced.

Clearer and more detailed introduction is needed on the taxonomy and existing phylogenetic understanding of the taxa being studied, and especially the section *Distegocarpus*. Is this the section *Distegocarpus* of Winkler (1904) or Li and Cheng (1979)? An authority is needed on line 32. The three species that make up this section need to be named early in the paper. The statement “sect. *Distegocarpus* ... exhibits intermediate morphology between these two genera and has an unsolved systematic position [10–13]” is inadequate. Reference 12 (Ma et al 2015) seems to be mis-cited as it does not include any species from *Distegocarpus*. Reference 11 (Yoo & Wen 2002) shows *Carpinus cordata* and *C. fangiana* (section *Distegocarpus*) as sister to a clade with the rest of *Carpinus* and *Ostrya* in an ITS analysis, but with *Carpinus japonica* (section *Distegocarpus*) as sister to them all. Reference 13 (Yang et al 2019) Shows *Ostrya* as sister to *Carpinus* in plastome tree but nested within *Carpinus* in an ITS tree. The authors need to cite Li JH. 2008 [Sequences of low-copy nuclear gene support the monophyly of *Ostrya* and paraphyly of *Carpinus* (Betulaceae). *Journal of Systematics and Evolution* 46: 333–340] whose analyses show *C. cordata* as sister to the rest of *Carpinus* plus *Ostrya* (with *C. japonica* grouping with the other *Carpinus* species.): thus, in this uncited study, Section *Distegocarpus* is not monophyletic. These existing phylogenetic treatments seem to suggest that *Ostrya* has arisen from within *Carpinus*. There does not seem to be an a priori reason to suspect from the existing analyses that Section *Distegocarpus* arose via hybridisation between *Ostrya* and *Carpinus*

sect. *Carpinus*. It is therefore unclear to me why the authors set up their study as a test of the hypothesis that *Distegocarpus* had a hybrid origin. It seems to me that this was a conclusion that arose from their phylogenomic study, rather than its starting hypothesis. To my mind the MS would be much clearer if it was written in this way: i.e. as a phylogenomic study on the unresolved relationships of *Carpinus* and *Ostrya*.

Response: Thanks for your detailed comments and helpful suggestions. According to your comments and suggestions, we carefully revised the manuscript on the following aspects: (1) we stated the taxonomy on taxa used in this study. The genus *Carpinus* L. was divided into two sections, sect. *Eucarpinus* Sarg. (= *Carpinus*) and sect. *Distegocarpus* (Sieb. et Zucc.) Sarg., by Winkler (1904). *C. fangiiana* Hu was added into the sect. *Distegocarpus* by Li and Cheng (1979). These two references were added in the manuscript. The authority of each species, section, and genus were all added when their name occurred for the first time; (2) we rephrased the introduction on taxa being studied. The three species from sect. *Distegocarpus* were named in this part; (3) we rephrased the introduction on the existing phylogenetic understanding of the taxa being studied. As the reviewer suggested, the primary Reference 12 (Ma et al., 2015) was removed and a new reference (Li JH, 2008) was inserted. We fully described and discussed the unsolved phylogenetic relationships of the genera *Ostrya* and *Carpinus*.

Actually, such phylogenetic discordance could not directly lead to the hypothesis of hybrid origin of sect. *Distegocarpus*. It was indeed a conclusion that arose from our subsequent phylogenomic study as you indicated. Considering the previous studies mainly based on the short sequences (e.g., ITS and *Nia*), which were susceptible to introgression, ILS, and HHS, we could not infer a firm conclusion about their evolutionary histories. We are so sorry that our primary statement is not clear enough. We have rephrased this part of description.

From a phylogenomic analysis, the authors provide very good evidence that the majority of genes support the monophyly of *Carpinus* sect. *Disegocarpus* and *Carpinus* sect. *Carpinus*, with *Ostrya* as a sister. They also give evidence for extremely high gene-tree incongruence among *Ostrya*, *Carpinus* sect. *Disegocarpus* and *Carpinus* sect. *Carpinus*. (Figure 1 b and c). These in themselves are interesting results and deserve to be highlighted more explicitly.

Response: Thanks for your helpful suggestions. We rearranged the figures and detailedly described our results of phylogenomic analysis. The new Fig. 1 illustrated the phylogenetic relationships of these taxa. We also explicitly highlighted the high gene-tree incongruence among *Ostrya*, *Carpinus* sect. *Disegocarpus*, and *Carpinus* sect. *Carpinus* in new Fig. 2. The new Fig. 2d and new Table S21 was added to describe the genome-wide distribution of these inconsistent gene trees.

The authors then seek to understand if this extreme gene-tree incongruence is due to incomplete lineage sorting (ILS) or introgressive hybridisation or homoploid hybrid speciation or convergent evolution (homoplasy). They claim to show that only homoploid hybrid speciation can explain the patterns they find.

To eliminate ILS, the authors assume that under ILS, similar numbers of genes would show the topologies II and III of Figure 1b. I can see how this would be true if all the genes were evolving in

a neutral fashion, or we were looking at a short timespan since the lineages split. However, we are looking at a timespan estimated to be around 30 million years, and the authors later present evidence for the action of biased selection (lines 91-92). I am not therefore convinced that they can eliminate ILS as a hypothesis.

Response: Thanks for your valuable question. Actually, these gene topologies were reconstructed using 4DTv (four-fold synonymous transversion) sites, which were a type of widely used neutral sites and capable of minimizing the interference due to biased selection (Zhang et al. 2017; Marburger et al. 2019). We added more detailed description in the Method and Results sections respectively. Furthermore, selection may cause the convergent mutations, a kind of homoplasy. In our subsequent population-level analyses, we fully described and discussed how to eliminate homoplasy. This was another important way to overcoming the effects of biased selection.

Zhang L, Su W, ... Kuang H. RNA sequencing provides insights into the evolution of lettuce and the regulation of flavonoid biosynthesis. Nature Communications. 2017. 8: 2264.

Marburger S, Monnahan P, ... Yant L. Interspecific introgression mediates adaptation to whole genome duplication. Nature Communications. 2019. 10: 5218.

To eliminate introgressive hybridisation the authors analyse genome sequences for many different species or individuals within *Ostrya*, *Carpinus* sect. *Disegocarpus* and *Carpinus* sect. *Carpinus*. They argue that if introgression had occurred since speciation occurred within these three groups, then different species within the three groups would show different levels of introgression. This argument is valid, but it does not eliminate the hypothesis of introgressive hybridisation occurring before the ancestor of the three species within *Carpinus* sect. *Disegocarpus* began to speciate. Even if all three species of *Carpinus* sect. *Disegocarpus* show similar levels of introgression from *Ostrya*, this could be due to introgression into their common ancestor. In other words, the ancestors of *Carpinus* sect. *Disegocarpus* and sect. *Carpinus* could have speciated before introgression from *Ostrya* into the ancestors of sect. *Disegocarpus* occurred. Thus, homoploid hybrid speciation between the ancestors of *Ostrya* and *Carpinus* sect. *Carpinus* is indistinguishable from introgressive hybridisation between *Ostrya* and an already diverged ancestor of *Carpinus* sect. *Disegocarpus*.

Response: Thanks for your valuable question. We have made great efforts on this question and really hope we can convince you. For a long term, the concepts of hybridization, introgression, and hybrid speciation have been confused. Hybridization is an intermittent or ongoing evolutionary event, which often lead to the introgression of adaptive genes (Taylor and Larson, 2019). Introgression (also named introgressive hybridization, including introgression between ancestors of different groups of taxa) and hybrid speciation are two aspects of hybridization (Taylor and Larson, 2019). ‘Hybrid speciation is defined as a speciation event in which hybridization has played a crucial role in the evolution of reproductive barriers between a hybrid lineage and its parent lineages’ (Taylor and Larson, 2019). In other word, when reproductive isolation (RI)-related loci are recombined after repeated recombination, the introgression will such lead to the hybrid speciation. So, introgressions (also called introgressive hybridization) with RI-recombination are the initial process of HHS and those without RI-recombination will merge with one parent via repeated backcrossing. As we claimed in the

manuscript, there is a major distinction between introgression and HHS based on population genetic data: introgression usually leads to genetic mixture in a few individuals or populations, rather than in all hybrid offspring as does HHS. This criterion was met by almost all studies on introgression or HHS based on population genomic data in recent years. Here, we provided a list of >120 papers about introgression and >20 papers on HHS (or homoploid hybrid origin) respectively, which were published on the top or famous specialized journals from 2012 to now (Table S1). The number of the papers on HHS seems to be fewer than that of introgression, because HHS is much rarer in nature. All these papers could support our claim. So, due to this major distinction between introgression and HHS, even if the introgression occurs between *Ostrya* and the ancestor of sect. *Disegocarpus*, it is very unlikely to observe the signals of introgression in all hybrid offspring. Our results of population-level genomic analyses could thus effectively eliminate the introgression between *Ostrya* and an already diverged ancestor of sect. *Disegocarpus*.

Under another scenario, sect. *Disegocarpus* diverged from sect. *Carpinus* firstly, and then hybridized with *Ostrya*. In this scenario, two possibilities might exist. First, some populations of sect. *Disegocarpus* were hybridized while the others not. The hybridized populations developed into one independent lineage to produce three species because all hybrid signals were detected. The non-hybridized populations should develop another lineage with close relationship with sect. *Carpinus*. However, we failed to find such a lineage. Therefore, this scenario is unlikely. Second, the ‘non-hybridized populations’ disappeared. Under this scenario, it is still fit for HHS. In fact, this scenario is basically same with the HHS assumption of our manuscript: some populations of the ancestor of sect. *Carpinus* were hybridized by the ancestor of *Ostrya* with further backcrossing (can be called ‘introgressive populations’), which further develop into a new and independent evolving lineage (therefore should be HHS) while those ‘non-introgressed populations’ developed the current sect. *Carpinus*.

According to the major distinction between HHS and introgression based on distributions of genetic admixture in the current populations, our multiple and integrated analyses may suggest whether one stable and independently evolving lineage had been established for HHS while intermittent and ongoing evolving for ‘introgression’, which may undergo further HHS or merge with one parent through repeated backcrossing. As stated before, we eliminated introgression (also called introgressive hybridization) on following aspects: (a) the application of population genomic data in ABBA-BABA test, HyDe analysis, and HHS test (a test for significant PIV signals) makes them capable of detecting hybrid signals with genetic information from all offspring and can therefore distinguish HHS from introgression; (b) the observed genome-wide evenly-distributed different gene phylogenies (Topologies I and II) and PIVs (a type of indels) from both assumed parental lineages (“CD” and “OD”) could also exclude it, because the introgression rarely led to even signals across the genome and (c) numerous alternative fixture of indels (PIVs in this study) from two parental lineages directly produced by hybridization, led to the genetic distinctions of the hybrid lineages from two parental lineages in numerous loci, which undoubtedly resulted in the postzygotic RI because of hybridization.

In the new submitted manuscript, we added the more detailed statement in the Introduction and Discussion sections respectively. We also added a new Fig. S1 to better illustrate the process

and the distinctions of HHS and introgression. In the new Fig. S1, for F1 hybrids (Fig. S1b), there are two possible evolutionary products after repeated recombination: (1) leading to introgressive populations in the first stage (Fig. S1c); (2) further leading to HHS (d) after a long independent evolution. For ‘introgressive populations’ (Fig. S1c), if the reproductive isolation (RI)-related loci are recombined (Fig. S1c, Scenario I), it may undergo HHS (Fig. S1d) after repeated recombination. We really hope our efforts can make you convinced.

[Fig. S1 | Homoploid hybrid speciation (HHS) and its distinction from other hybridization outcomes. a, Divergence of ancient species. b, F1 hybrids. c, Introgression. d, HHS (arising from both F1 or backcrossing hybrids with equal or unequal genomic contributions from two parents). In (c), introgression comprises two scenarios: recombination of RI-related loci (Scenario I, also called introgressive hybridization) and recombination of non-RI-related loci (Scenario II). Scenario I can be also defined the initial HHS process.]

Taylor SA, Larson EL. Insights from genomes into the evolutionary importance and prevalence of hybridization in nature. *Nature Ecology and Evolution*. 2019. 3: 170–177.

In any case, I do not find Figure 1f (see also line 68) clear evidence against introgression since the divergence of sect *Disegocarpus*, as even if some recent introgression had occurred, this topology could have been found, based on genome-wide SNPS (line 177). The ABBA-BABA test, HyDe and HHS tests seem to provide more compelling evidence that hybridisation occurred prior to the divergence of the species within section *Disegocarpus* (though the methods here need to be explained more clearly). But they do not (as far as I can see) distinguish between homoploid hybrid speciation and introgressive hybridisation between *Ostrya* and an already diverged ancestor of *Carpinus* sect *Disegocarpus*.

Response: Thanks for your good question. Actually, the red dotted arrow was mainly used to exhibit the results of HyDe analysis, in which sect. *Disegocarpus* was inferred as a homoploid hybrid lineage from *Ostrya* and sect. *Carpinus*. We are sorry for making you confused. We have rearranged the figures. A new Fig. 3 was used to show the results of ABBA-BABA test and HyDe analysis. For the latter question – how to distinguish between HHS and introgressive hybridisation between *Ostrya* and an already diverged ancestor of *Carpinus* sect. *Disegocarpus*, we hope our previous response can convince you. Our population-level genomic analyses could effectively distinguish HHS from the introgression between *Ostrya* and an already diverged ancestor of *Carpinus* sect. *Disegocarpus*. Especially for the HHS test (a test for significant PIV signals), it was based on the population-level indels which were fixed between different lineages.

HyDe was one of the most widely used approach to infer HHS events, which could infer the homoploid hybrid lineage and two putative parental lineages (Kong and Kubatka, 2021). Its algorithm (models) for population-level data could eliminate ILS and homoplasy (Blischak et al. 2018). ABBA-BABA test was also a widely used method to provide corroborative evidence for HHS (Kong and Kubatka, 2021), for which could indicate the gene flow with the exclusion of ILS. The application of population genomic data in ABBA-BABA test and HyDe analysis makes them capable of detecting hybrid signals with genetic information from all offspring and can therefore distinguish HHS from introgression. We confirmed the HHS event by such an integrated set of methods. We have expanded our manuscript and explained it more clearly.

Blischak PD, Chifman J, Wolfe AD, Kubatko LS. HyDe: a Python package for genome-scale hybridization detection. Systematic Biology. 2018. 67: 821–829.

Kong S, Kubatko LS. Comparative performance of popular methods for hybrid detection using genomic data. Systematic Biology. 2021. 70: 891–907.

The hypothesis of convergent evolution is given less attention by the authors. They seek to deal with this using long indels (line 72) as markers in the hybridisation tests. They could say more about why they expect these to be unlikely to have evolved convergently. But overall, I agree with them that convergent evolution is very unlikely to underlie the gene tree incongruence that they find.

Response: Thanks for your approval and suggestion. We have added the related statement in the Results and Discussion sections respectively.

The final section of the MS (lines 86 to 114) is built on the assumption that the authors have conclusively demonstrated that homoploid hybrid speciation has occurred. As I am unconvinced that they have demonstrated this, this section does not convince me either. Some of the patterns they report on genes under selection are nonetheless interesting – I would just like to see them discussed in a framework of gene tree incongruence, rather than homoploid hybrid speciation.

Response: Thanks for your good comments. We have made great efforts on confirming HHS scenario: from the distinction of their concepts and genetic features to a list of ~ 150 published good papers (Table S1) supporting our perspective, and from our methodological principles to a more detailed discussion. We hope we can convince you this time. We also highlighted the gene

tree incongruence as you suggested, including the rearrangement of figures, adding a new Fig. 2d and a new Table S21, and a more detailed description in the Results section.

There are numerous linguistic infelicities throughout the paper that also need to be worked on.

Response: Thanks for your suggestions. We are sorry for our poor English. The manuscript has been edited by a native English speaker.

The authors may also wish to consult these two papers:

Guo, Xing, Daniel C. Thomas, and Richard MK Saunders. "Gene tree discordance and coalescent methods support ancient intergeneric hybridisation between *Dasymaschalon* and *Friesodielsia* (Annonaceae)." *Molecular Phylogenetics and Evolution* 127 (2018): 14-29. This seems to be a somewhat similar paper to the present MS.

Crane, P. R. "Betulaceous leaves and fruits from the British Upper Palaeocene." *Botanical Journal of the Linnean Society* 83.2 (1981): 103-136. This reports a fossil that shows a "combination of the characters of several living genera, particularly *Corylus* and *Carpinus*"

Response: Thanks for your helpful recommendations. We have carefully read these papers and cited them in the manuscript.

Reviewer #2 (Remarks to the Author):

This paper analyses genome level data from multiple tree species in two genera and identify a clade that has originated through hybridization between genera. While studies confirming homoploid hybrid speciation are becoming more frequent thanks to the resolution provided by genomic data, the novelty of this study is to detect an older hybridization event predating generic divergence. This is an advance of interest.

The approach using orthologous gene sets and indels seem robust to detect hybridization and distinguish it from incomplete lineage sorting and gene flow. The new reference *Caprinus* genome is built from extensive sequence data and is likely to be highly reliable. The inclusion of an improved *Ostrya* genome is valuable to prevent ascertainment biases influencing subsequent analyses. However, later in the methods, I'm disappointed to learn that population samples were mapped against just one of the available genomes, missing an opportunity to protect against ascertainment bias. The authors should justify this approach.

The hypothesis to distinguish HHS from introgression needs to be better explained as it is vital for the reader to assess the subsequent results and interpretations. Also tests such as HyDe are presented to support HHS over introgression without sufficient information for the reader about how this distinction can be made.

The discussion about selection in the new hybrid lineage is interesting but the results as presented have been over-interpreted. I understand the evidence to be that some genes are alternately fixed in the hybrid lineage. This is not sufficient evidence in itself for either selection or reproductive incompatibilities.

The language used is awkward but mostly understandable so it would be worth having the paper edited by a native language speaker.

Overall, this is an exciting study system to demonstrate an older HHS event but the paper needs to present stronger evidence of HHS versus introgression to be really convincing. This could simply be a case of carefully explaining the tested hypotheses and the logic of the tests applied. Not all researchers in the field argue for such a strong distinction to be made between introgression and HHS, but simply demonstrating past introgression as part of diversification would be less novel in my opinion.

Response: Thanks for your positive comments and helpful suggestions. Actually, the primary manuscript was in the traditional 'Short Report' format, which may be too short to provide enough information. We have expanded the manuscript to a full-length paper format and revised the manuscript according to your suggestions.

Specific comments

L11-13 Indels incorrectly capitalized here and elsewhere. I would replace "of" with "from" here. I'm not sure how indels data can exclude hypotheses of introgression.

Response: Thanks for your suggestions. We corrected the words according to your suggestions. Actually, hypothesis of introgression cannot be simply excluded by the data type of indels, but due to the applied integrated methods based on population genomic data, e.g., HyDe analysis, ABBA-BABA test, HHS test (a modified test for significant PIV signals), especially for the application of population genomic data. We changed the expression here and explained how to exclude introgression hypothesis in detail in the new submitted manuscript.

L32 State the genus that sect. *Distegocarpus* is currently assigned to, or if there is controversy. It is important to clearly explain the background to the system to help readers evaluate the advances of the study.

Response: We have revised the manuscript according to your valuable suggestions.

L40 It would be worth stating the Hi-C sequencing coverage of the *O. rehderiana* genome here for comparison with the *C. viminea* genome.

Response: We have added the information according to your suggestion.

L55-56 Mention and justify mutation rate and generation time used to estimate divergence times.

Response: Thanks for your good question. The three lineages have diverged for a long term. It is quite difficult to estimate the accurate time of divergence for the lineages undergoing complex

evolutionary history (e.g., HHS and ILS et al.). So, we roughly dated the times of divergence between the three lineages from *Ks* distributions of the representative species (*C. viminea*, *C. fangiana*, *O. rehderiana*, and *Betula pendula* as outgroup) under a fossil constraint. According to our formula transformation, the accurate information of mutation rate and generation time was not needed. Based on the formula: $T=Ks/2\mu$ (where *T* indicates the time of divergence between two species, *Ks* indicates the *Ks* peak value, and μ indicates the mutation rate), it follows that: $T_1/T_2=(Ks_1/2\mu_1)/(Ks_2/2\mu_2)$. Supposing that different lineages have similar mutation rates, we have: $T_1/T_2=Ks_1/Ks_2$. Based on previous studies, the divergence time between *Betula pendula* and the other three species was set as ~71 million years ago (mya). From this, the genera *Ostrya* and *Carpinus* were estimated to have diverged at 23-33 mya and the divergence time between sects. *Carpinus* and *Distegocarpus* was dated to 17-26 mya. We modified the description of this method in the Main Text. The detailed method was stated clearly in the Supplementary Information.

L56-58 Why is no divergence time between *C. fangiana* and *O. rehderiana* provided?

Response: Thanks for your question. We are sorry for making you confused. As our response for previous question, we roughly dated the divergence times of the three lineages from *Ks* distributions of the representative species (*C. viminea*, *C. fangiana*, *O. rehderiana*, and *Betula pendula* as outgroup). According to the phylogeny, the divergence time between sect. *Distegocarpus* (representative by *C. fangiana*) and *Ostrya* (representative by *O. rehderiana*) should be equal to that between sect. *Carpinus* (representative by *C. viminea*) and *Ostrya*. So, it was not provided. We have revised the statement of this part to demonstrate it clearly.

Figure 1 What species is represented by *Bpe* in panel e and why is it important to show it in this panel? The HyDe summary is too vague. State what is being quantified here.

Response: *Bpe* (*Betula pendula*) was used as the outgroup to date the times of divergence between it and *Ostrya rehderiana* (*Ore*), *Carpinus viminea* (*Cvi*), and *C. fangiana* (*Cfa*). Based on previous studies, the divergence time between *Betula pendula* and the other three species was set as ~71 million years ago according to a fossil constraint. Based on our formula transformation, we could thus estimate the times of divergence between the three lineages. The detailed information was added in the figure legend, Main Text, and Supplementary Information. We have described the HyDe summary in detail. A new Fig. 3b was added to show the results of HyDe analysis.

L59-60 Why would not all hybrid offspring show signals of admixture? This hypothesis needs to be really clear.

Response: Thanks for your helpful suggestion. Actually, whether all hybrid offspring show signals of admixture is a major distinction between introgression and HHS. We have made great efforts on claiming this point. More details were described in the Introduction and Discussion. To better illustrate it, we also added a new Figure S1 and a list of ~ 150 related studies (Table S1). These studies were focused on introgression and/ or HHS and based on population genomic data. They covered almost all relevant studies published on the top or famous specialized journals from 2012 to now. All of these studies satisfied this criterion.

L66 Which reference was used and why?

Response: Our new generated assembly of *C. viminea* was used as the reference genome. We selected it as the reference genome for the following two aspects: (1) *C. viminea* genome was sequenced by Nanopore long reads and had a much better continuity (Contig N50 = 4.31 Mb), while the genome of *O. rehderiana* was sequenced with Illumina short reads (by Illumina pair-end libraries and Illumina mate-pair libraries) and had a Contig N50 of only 22 Kb; (2) we also randomly selected 16 individuals (including 4 samples from sect. *Carpinus*, 8 samples from sect. *Distegocarpus*, and 4 samples from *Ostrya*) and mapped them to both genomes respectively. We compared their mapped site number, reads mapping ratio, and average mapping depth for different reference genomes. Obviously, when *C. viminea* was used as the reference genome, most individuals would show a better mapping quality, because more individuals sequenced in this study (37 individuals from sects. *Carpinus* and *Distegocarpus* versus 7 individuals from *Ostrya*) had a closer phylogenetic relationship with *C. viminea*. Actually, according to plenty of previous studies with population genomic data, we convinced different reference genomes would not affect our results. We have added this information in the Main Text and Supplementary Information. We also added a new Table S19 to illustrate it.

L71 I think you should briefly mention the lineages used for each of the ABBA-BABA and HyDe tests here.

Response: Thanks for your helpful suggestion. We have rearranged the figures and added a new Fig. 3 to illustrate the results of ABBA-BABA and HyDe test. The information of the used lineages was shown in the figure.

L72 The phrase "at utmost" is not clear. I think you are arguing that indels are effective at excluding the hypothesis of homoplasy.

Response: Thanks for your comment. We have revised this sentence.

L74-75 Briefly explain how HyDe can distinguish HHS from introgression.

Response: Thanks for your helpful suggestion. HyDe could distinguish HHS from introgression when performing with population genomic data (Blischak et al. 2018), for being capable of detecting hybrid signals with genetic information from all offspring and can therefore distinguish HHS from introgression. We revised the manuscript as your suggestion.

Blischak PD, Chifman J, Wolfe AD, Kubatko LS. HyDe: a Python package for genome-scale hybridization detection. Systematic Biology. 2018. 67: 821–829.

L87-89 Not all alleles will alternately fix due to selection on reproductive incompatibilities. Drift could play a role, as could directional selection on new hybrids.

Response: Thanks for your good question. We appreciated on your opinion. Of course, not all alleles will alternately be fixed due to selection on reproductive incompatibilities. However, for

HHS lineage, not only phylogenies produced by a sliding-window strategy but also diverged alleles and indels, all of them would form a typical “mosaic model”. The alleles derived from different parental lineages could not be exactly alternately fixed, but on the whole they would be evenly-distributed at genome-wide scale (Wang et al. 2021). This alternative fixture will lead to the initial postzygotic RIs. Yes, you are right that directional selection on the new hybrids and drifts may further lead to more RIs. However, this occurs after the initial establishment of one independently evolving lineage. We added one sentence in the Discussion section: *With these initial RIs created by hybridization, the total RI between sect. Distegocarpus and each parental lineage could have been continuously accumulated and enforced along its independent evolution through drift and directional selection.*

Wang ZF, Jiang YZ, ..., Liu JQ. Hybrid speciation via inheritance of alternate alleles of parental isolating genes. *Molecular Plant*. 2021. 14: 208–222.

L90 Similar to my previous point, the term positively selected genes would only be valid if there is a specific test of selection over and above being fixed in the new hybrid lineage. State the test applied or change this term to alternately fixed genes.

Response: Thanks for your good question. We have added the detailed information about the test we applied.

L99 I don't follow the distinction between mutations and alleles. From the following example, i think that you are referring to a combination of alternate mutations from each parental lineage within the same gene. Clarify the explanation here.

Response: Thanks for your helpful comment. We have revised this sentence according to your suggestion.

L105 Gene now named "LUC" rather than "LUG", one of these is a typo.

Response: We have corrected it.

L135 Typo, change "predicated" to "predicted".

Response: We have corrected it.

L161 Typo, change "almost samples" to "almost all samples"

Response: We have corrected it.

L165-167 What was the reason for mapping all samples to *C. viminea*, when there was also the choice of the *O. rehderiana* genome? Can you justify that this would not lead to ascertainment bias in SNP calling?

Response: Thanks for your valuable question. Population genomic analysis is based on the information of shared variants between all used samples. So, for most studies, all samples must

map to one reference genome. Reads mapping and variants calling would cost large amount of computing time. It is really difficult for us to map all data to another genome and compare their difference. According to plenty of previous studies with population genomic data, we convinced different reference genomes would not affect our results. Actually, most previous studies were also based on only one reference genome. So, we selected *C. viminea* as the reference genome according to its much higher continuity and better mapping quality for more samples as we previously stated.

L182-186 I recommend adding a little more detail on the logic of this relatively new test here about how it can distinguish between introgression and HHS.

Response: Thanks for your helpful suggestion. We have added a more detailed description.

L195-197 The last part of this methods needs to be rewritten for clarity.

Response: Thanks for your helpful suggestion. We have revised this part.

Figure 2 The consistent lineage colours from Figure 1 are useful but the changed naming system is confusing. I prefer the C,D,O,A naming system of figure 2. Indicate which genome is shown in panel a, and which rows are PIVs and AVs in panel b. Panel c seems to show the crucial test of HHS. Explain this test in the main text.

Response: Thanks for your helpful suggestion. We tried to use the consistent lineage colors and naming system to make the manuscript more readable. For many readers not familiar with our study system, we tried to use detailed name of each lineage. However, for different combinations of lineages in old Fig. 2a-c, it was really difficult to show the full name due to the limited space and appearance, so we used “CDOA” naming system here. Now, we have rearranged the figures to exhibit only one naming system in one figure. The “CDOA” naming system only occurred in new Fig. 4 now. Panel a exhibited the PIVs identified based on population genomic data (from all the three lineages). We added related information in the figure legend as you suggested. A more detailed information was described in the Note section of figure legend. The explanation of HHS test was also added in the main text. A more detailed explanation was in the Supplementary Information.

Reviewer #3 (Remarks to the Author):

This paper looked at the evolution of a plant family to argue it is a product of homoploid hybrid speciation (HHS). I disagree with the authors claims and suggest a simple scenario where there's been hybridization between sect *Distegocarpus* (which the authors claim are the hybrid) and *Ostrya* (the third outgroup), without invoking any evidence of speciation is sufficient to explain all their results. And based on this result one does not have to invoke HHS at all for their study system. I'd suggest the authors to read the review from Schumer et al. (<https://onlinelibrary.wiley.com/doi/10.1111/evo.12399>) for details on what HHS is, since simply

having evidence of hybridization is not sufficient to invoke hybrid speciation.

Response: We really appreciate your constructive comments and valuable suggestions. Actually, the primary manuscript was written in the traditional ‘Short Report’ format, which may be too short to provide enough details. We are so sorry that our manuscript could not convince you. Here, we revised the manuscript and expanded the revised manuscript to the full-length article format to demonstrate more details.

In the new submitted manuscript, we have made great efforts on your first question - whether to have another scenario where there's been hybridization between sect *Distegocarpus* and *Ostrya* (an introgression from secondary contact between the two lineages). For a long term, the concepts of hybridization, introgression, and hybrid speciation have been confused. Hybridization is an intermittent or ongoing evolutionary event, which often lead to the introgression of adaptive genes (Taylor and Larson, 2019). Introgression (also named introgressive hybridization, including introgression between ancestors of different groups of taxa) and hybrid speciation are two outcomes of hybridization (Taylor and Larson, 2019). ‘Hybrid speciation is defined as a speciation event in which hybridization has played a crucial role in the evolution of reproductive barriers between a hybrid lineage and its parent lineages’ (Taylor and Larson, 2019). When reproductive isolation (RI)-related loci are recombined via repeated recombination, the introgression will such lead to HHS. So, introgressions (also called introgressive hybridization) with RI-recombination are the initial process of HHS and those without RI- recombination will merge with one parent via repeated backcrossing. In the new submitted manuscript, we added the more detailed statement in the Introduction and Discussion sections respectively. We also added a new Fig. S1 to better illustrate the process and the distinctions of HHS and introgression.

[Fig. S1 | Homoploid hybrid speciation (HHS) and its distinction from other hybridization outcomes. a, Divergence of ancient species. b, F1 hybrids. c, Introgression. d, HHS (arising from

both F1 or backcrossing hybrids with equal or unequal genomic contributions from two parents). In (c), introgression comprises two scenarios: recombination of RI-related loci (Scenario I, also called introgressive hybridization) and recombination of non-RI-related loci (Scenario II). Scenario I can be also defined the initial HHS process.]

As we claimed in the manuscript, there is a major distinction between introgression and HHS: introgression usually leads to genetic mixture in a few individuals or populations, rather than in all hybrid offspring as does HHS. This criterion was met by almost all studies on introgression or HHS based on population genomic data in recent years. Here, we provided a list of >120 papers about introgression and >20 papers on HHS (or homoploid hybrid origin) respectively. These papers covered almost all relevant studies published on the top or famous specialized journals from 2012 to now (Table S1). The number of the papers on HHS seems to be fewer than that of introgression, because HHS is much rarer in nature. All these papers satisfied this criterion and thus could support our perspective.

Due to this major distinction between introgression and HHS, we can thus eliminate the introgression scenario (hybridization between sect *Distegocarpus* and *Ostrya*) on following aspects: (a) the application of population genomic data in ABBA-BABA test, HyDe analysis, and HHS test (a test for significant PIV signals) makes them capable of detecting hybrid signals with genetic information from all offspring and can therefore distinguish HHS from introgression; (b) the observed genome-wide evenly-distributed different gene phylogenies (Topologies I and II) and PIVs (a type of indels) from both assumed parental lineages (“CD” and “OD”) could also exclude it, because the introgression rarely led to evenly signals across the genome and (c) numerous alternative fixture of indels (PIVs in this study) from two parental lineages directly produced by hybridization, led to the genetic distinctions of the hybrid lineages from two parental lineages in numerous loci, which undoubtedly resulted in the postzygotic RI because of hybridization. We rewrote the manuscript for illustrating more details on this question. Much detailed description was added in Introduction, Results, Discussion, and Methods sections respectively.

We really appreciated your second comment - the criteria for HHS. We carefully read the paper you recommended (Schumer et al. 2014; actually we have been very familiar with this paper for a long time). According to Schumer et al. (2014), “To demonstrate that hybrid speciation has occurred given this definition, three criteria must be satisfied: (1) reproductive isolation of hybrid lineages from the parental species, (2) evidence of hybridization in the genome, and (3) evidence that this reproductive isolation is a consequence of hybridization.” Our results met each of these criteria respectively:

- (a) The three ancient diverged lineages in this study were well defined. They have diverged for a long term (17-33 mya) and further produced multiple well-defined daughter species. The two parental lineages (*Ostrya* and sect. *Carpinus*) belong to different genera. Sect. *Distegocarpus* (the hybrid lineage) and sect. *Carpinus* belong different sections. Reproductive isolation (RI) obviously existed between them.
- (b) All of our analyses provided the evidence of hybridization, including the much discordant

gene phylogenies, a simulation to eliminate effects of incomplete lineage sorting (ILS), the ABBA-BABA test and HyDe analysis based on population genomic data, the HHS test (a test for significant PIV signals), the genome-wide evenly-distributed discordant gene topologies and PIV signals (a type of indels), the detected PSGs of sect. *Distegocarpus* derived from both parental lineages, and also the hybrid signals in certain genes. In fact, all long indels found here in Sect. *Distegocarpus* were found to be specific to sect. *Carpinus* or *Ostrya* across the total genome, indicating the genomic admixture of Sect. *Distegocarpus* from the other two groups.

- (c) We provided the evidence that hybridization event led to RI (not completely) of sect. *Distegocarpus* from two parental lineages. PSGs identified in sect. *Distegocarpus* were found to derive from sect. *Carpinus* and *Ostrya* respectively, and functional analyses suggested that some of them are involved in flowering time and environmental adaptation. Sect. *Distegocarpus* therefore inherited prezygotic RI genes alternately from the two parental lineages, which would have immediately contributed to its RI from each of the two parental lineages at the initial HHS stage through differences in flowering time and habitat adaptation. In addition, all alternative fixed indels from ancestral lineages (PIVs in this study) across numerous loci may have further contributed to immediate post-zygotic RI according to HHS modelling (Schumer et al. 2015). Thus these findings overall suggest that genetic admixtures in sect. *Distegocarpus* due to hybridization could have directly produced RI and therefore three strict criteria of the HHS hypothesis are well met (Schumer et al. 2014).

The related description and discussion were added in the new submitted manuscript. We really hope we can convince you.

Schumer M, Rosenthal GG, Andolfatto P. How common is homoploid hybrid speciation? *Evolution*. 2014. 68: 1553–1560.

Schumer M, Cui R, Rosenthal GG, Andolfatto P. Reproductive isolation of hybrid populations driven by genetic incompatibilities. *Plos Genetics*. 2015. 11: e1005041.

Taylor SA, Larson EL. Insights from genomes into the evolutionary importance and prevalence of hybridization in nature. *Nature Ecology and Evolution*. 2019. 3: 170–177.

Now I agree that the authors have evidence of a hybridization. Although I think some of the results are basically redundant to each other, for instance Fig 1C and Fig 2A,B is basically saying among three possible topologies for a rooted three taxon group, between the two minor topologies one is significantly more abundant than the other. I agree this is evidence of hybridization but not sure it warrants a separate figure for the results. But importantly still how can the authors eliminate the possibility that this is simply a hybridization from secondary contact? For this reason I think the evidence that this system is HHS is very weak.

Response: Thanks for your good question. Actually, Fig. 1c and Fig. 2a,b were two distinct analyses based on different types of data. Fig. 1c (new Fig. 2 in the revised manuscript) showed the much incongruent gene phylogenies based on *de novo* genome sequences. This was evidence

of hybridization. Meanwhile, the genome-wide evenly-distributed incongruent topologies were another evidence for hybridization, because introgression could rarely lead to such evenly distribution. We highlighted this part of description as the suggestions from Reviewer #1 and added a new Fig. 2d. However, Fig. 2a,b (new Fig. 4 in the revised manuscript) illustrated the genome-wide evenly-distributed of PIVs (a type of indels) identified based on the population genomic data. We modified the approach developed by Jiang et al. (2020) and applied it to the population-level data. The relative long indels (> 5 bp) fixed between different lineages were used in our analyses. Ancestral variations (AVs) and phylogenetically informative variations (PIVs) were identified across the genome and we further detected the significant PIV signals. AVs and PIVs were classified based on their times of occurrence (Jiang et al. 2020). AVs occurred before the differentiation of all species. PIVs occurred after the first species differentiated and before the last one. If significant PIV signals can be detected between the assumed hybrid species and each parental species, the HHS assumption would be validated with elimination of introgression and ILS. The application of population genomic data could make it capable of detecting hybrid signals in all offspring, and thus eliminate the introgression. The genome-wide evenly distribution of PIVs also could be as evidence for HHS. For your second comment on a hybridization from secondary contact (a kind of a hybridization from secondary contact), we really hope our previous response and efforts on revision could convince you.

Jiang YF, Yuan ZW, ..., Liu CJ. Differentiating homoploid hybridization from ancestral subdivision in evaluating the origin of the D lineage in wheat. *New Phytologist*. 2020. 228: 409–414.

Also they conduct evidence of positive selection, but the method they use (an HKA test) is very underwhelming. There are so many powerful tests that look for evidence of a selective sweep genome-wide and the lack of using any of these state-of-the art methods is a concern. In addition, anybody really can make a story out of false positive selective sweep results (see <https://academic.oup.com/mbe/article/29/10/3237/1032149>) that their positive selection results dont seem that convincing.

Response: Thanks for your good question. We detected the PSGs in the hybrid lineage (which were shared and derived from each parental lineage) by our previously reported method (Wang et al. 2021). The paper containing this method was primarily published in *Molecular Plant* as a cover story (Wang et al. 2021). Recently, it has been awarded as the ‘Top 10 high impact papers on genomics and evolutionary biology in 2020 and 2021’ by the Editorial Board of *Molecular Plant*. According to the data of Google Scholar, it has been cited for 11 times in the past half year, soon after its publication. Actually, it was not a simple HKA test, but an integrated pipeline comprising of a well-known HKA test, divergence assessments for nonsynonymous mutations, and phylogenetic analyses.

We appreciated genomic analyses might lead to false positive results. Functional tests may be the “gold standard” to validate RI caused by such PSGs. But it was really difficult to perform functional tests on such non-model plants due to technical limitations and time cost. To our knowledge, the method we applied was the only pipeline hitherto developed to detect PSGs in HHS lineages and has been verified as effective by transgenic and common garden experiments (Wang et al. 2021). Actually, besides in Wang et al. (2021), the RI-related genes detected by this

method have been functionally validated in many other HHS lineages (e.g., a primate species and two independent HHS analyses on different species in *Rhododendron* genus), even though the papers have not been published yet (one is under review in *Science* for four month and the other two are in preparation now). We have described more details of this method in the new submitted manuscript.

Wang ZF, Jiang YZ, ..., Liu JQ. Hybrid speciation via inheritance of alternate alleles of parental isolating genes. *Molecular Plant*. 2021. 14: 208–222.

In sum, I think it requires more evidence then presented in this manuscript to show a certain lineage is a product of HHS. One that includes showing reproductive isolation and evolutionary genetic analysis that shows hybridization of the parental taxa, and not evidence of a hybrid swarm or secundary contact. But even if I give all the benefit of doubt for the authors, I think in the end the bottomline of this manuscript simply is that there is a lineage and its a product of HHS, which is a pretty limited scope in my view.

Response: Thanks for your good question. We are so sorry that, the primary manuscript was written in the traditional ‘Short Report’ format, which may be too short to provide enough details. Here, we revised and expanded the revised manuscript to the full-length article format to demonstrate more details. We really hope we can convince you.

We introduced more detailed information on the taxonomy of our study lineages. he two parental lineages (*Ostrya* and sect. *Carpinus*) belong to different genera. Sect. *Distegocarpus* (the hybrid lineage) and sect. *Carpinus* belong to different sections. The three ancient diverged lineages were well defined. They have diverged for a long term (17-33 mya) and further produced multiple well-defined daughter species. So, reproductive isolation (RI) obviously existed between them and they could not be a hybrid swarm. We also hope our previous response for how to distinguish HHS from introgression (including those produced by secondary contact) could convince you.

We think this manuscript is not only a case. Compared to the previous works, it is markedly strengthened in both significant discoveries and the ground-breaking novelty methods. We strongly believe that our work represents by far the most convincing demonstration (actually also the only one report) for HHS at higher taxonomic levels. Our confidence is based on the following aspects:

(a) For the significant discoveries: With integrated methods (including the comparative genomic analyses and population genomic analyses), we revealed that *Carpinus* sect. *Distegocarpus* with three species originated through HHS during the early divergence between two different genera, *Carpinus* and *Ostrya*. After the origin, the hybrid lineage and its two parents further produce multiple daughter species. To our knowledge, this is the first (also the only one) intergeneric HHS case with the most ancient time scale. Such an initial HHS event before further species diversification may lead to inconsistent gene tree topologies and reticulate phylogenies widely reported in plants at higher taxonomic levels. We also identified the candidate RI-related genes. These findings overall suggest that genetic admixtures in sect. *Distegocarpus* due to hybridization could have directly produced RI and

therefore the strict criteria of the HHS hypothesis are met. In addition, the genes contained alternative amino acid mutations fixed by each of the two parental lineages were identified. These recombinant genes due to hybridization may together have led to the intermediate nutlet bracts in sect. *Distegocarpus*. The genetic basis for such a new phenotype created by hybrid speciation at the high taxonomic level were revealed for the first time.

- (b) For the significant methodological advances: The traditional methods for detecting HHS are mainly based on the SNPs. For the random and/or convergent nucleotide mutations during the long-term evolutionary histories, it really hard to detect the ancient HHS event. The long indels is reported to be effective in excluding most such evolutionary homoplasy. However, the only one method based on indels (Jiang et al. 2020) could only apply on the single genomes, which might yield misleading results caused by introgression in a few individuals rather than all offspring. To avoid such errors, we first applied the population-level indel's information in the HHS analyses. We performed the ABBA-BABA statistics and HyDe analysis using the population-level indels. We also modified the method developed by Jiang et al. (2020) and applied it to the population-level data. It seems maybe not very complex or not heavy workload, but it is a pioneering work due to its innovation. We believe that our study will attract those researchers to detect the ancient HHS events in their focused groups, and will be cited frequently by the numerous research groups in the world. These reasons will also bring in a broad readership.
- (c) For the broad impact on science: Since Darwin, the origin of species has been attracting numerous researchers and the general public. We believe that our first demonstration of HHS at the genus level and also genetic basis for the new phenotype because of hybrid recombination will arouse widespread interest and cited frequently. This is not an addition of HHS case. However, this high taxonomic level could explain the predominant net evolution revealed by phylogenomic analyses in the world.

Jiang YF, Yuan ZW, ..., Liu CJ. Differentiating homoploid hybridization from ancestral subdivision in evaluating the origin of the D lineage in wheat. *New Phytologist*. 2020. 228: 409–414.

I'd also suggest the authors could give biological background on their study system. The lack of any background also makes it hard as a reader to understand why this system is a potential case of HHS, and understand (or even sympathize) the authors case to present this system as a case of HHS. In fact I didn't even understand if this was an animal or plant system until I saw Fig3 plant cartoons.

Response: Thanks for your valuable suggestion. We are so sorry for your confusion. We have rewritten and expanded the manuscript to a full-length paper format. The biological background on our study system was described in detail as you suggested. We also added more detailed information on HHS.

Reviewer #4 (Remarks to the Author):

This manuscript describes the collection and curation of data that are then used to examine the question of ancient hybridization between distinct plant genera leading to formation of a new species. A variety of methods are applied to the genomic data: a simulation-based test based on tree topologies; the ABBA-BABA test; HyDe; and a more recently proposed test based on phylogenetic informative variations. The data sets developed and methods applied are interesting, and the question is one that is commonly asked with such data in plants. In this regard, the manuscript would appeal to a broad audience.

However, the manuscript at present is poorly written and difficult to read. In addition to many grammatical errors, there is not enough information given with regard to the methods used to test for hybridization. There's the potential these have been applied correctly, but it would be impossible for the reader to know and, most importantly, impossible for a reader to replicate and apply to their own data. I provide some specific examples below. Minor typos and grammatical errors are listed below that for the main text; other sections contain similar errors, but I didn't list them all.

Response: Thanks for your positive comments. We are very sorry that some details were not illustrated clearly. Actually, the primary manuscript was written in the traditional "Short Report" format, which may be too short to provide enough information. We have expanded the manuscript to a full-length paper format. The manuscript has been revised according to your helpful suggestions. The language has been edited by a native English speaker.

Major issues:

1 — I don't understand why a simulation approach is being used (line 52). These are three-tip trees, right? The coalescent model specifies the probabilities of the three topologies in the absence of hybridization: the gene tree matching the species tree has probability $1 - \frac{2}{3} \exp(-t)$ and the other two topologies have probabilities $\frac{1}{3} \exp(-t)$, where t is the length of the single internal branch in coalescent units. Why simulate when these probabilities could just be calculated? And, whether simulation or exact calculation is to be used, how is t chosen? This needs to be specified in the manuscript, as it will significantly impact the results.

Response: Thanks for your good question. In our study, we used RAxML to construct the maximum likelihood (ML) tree for each group of orthologous genes. Then ASTRAL was used to estimate the species tree under a multi-species coalescent model. The branch lengths of the species tree so generated were in coalescent units. DendroPy was then applied to simulate the gene trees under the effects of incomplete lineage sorting (ILS). For DendroPy, we need not assign a t value, but used the previously produced coalescent-based species tree as the input directly. We think the t value was produced by ASTRAL when generating the species tree, and then transferred into DendroPy in the form of tree file. We have described more details of this simulation process in the main text.

According to the formulas you indicated, we could obtain the expected values of the three topologies, where the numbers of Topology II and Topology III would be expected to be equal (as we indicated in the main text). However, we observed the number of Topology II was more than that of Topology III. So, we then performed the simulation. The main purpose was to

examine whether significant differences were between them. In other word, we wanted to examine whether our conclusions had a statistical significance. We have added the related statement in the main text.

2—I think the authors spend too much time discussing homoplasy, given that the methods they use (e.g., HyDe) can handle homoplasy. Model-based methods like HyDe include the possibility of homoplasy as part of the substitution model. As an example, I don't understand the sentence (lines 28-29) “The well-assembled genomes and the accordingly extracted long [I]ndels can exclude most of such evolutionary homoplasy.” How do they “exclude” it? Also, why is an indel of more than 5bp considered long? This seems quite short in terms of phylogenetic information.

Response: Thanks for your good question. We appreciate your approval on the availability of HyDe. Actually, inaccurate genome sequences (those with inaccurate single-base and indels) always lead to the wrong phylogenies, some of which were due to the false positive of homoplasy (such as convergent mutations). With the rapidly growing of genomics, the quality of genomes is becoming much higher than previous. The well-assembled genomes with more accurate sequences will minimize such false positive. In addition, a high-quality genome (especially for those well-assembled with high continuity) will be expected to obtain more accurate information of indels from population genomic data.

“Homoplasy is a general term denoting that the acquisition of the same character state in two taxa is not because of common descent.”, which can arise by parallel/ convergent evolution or secondary loss (Rokas & Holland, 2000). According to previous studies, indels are relatively rare type of genomic events with much lower frequency of homoplasy than SNPs (Rokas & Holland, 2000; Bapteste & Philippe, 2002). Long indels is well known for a less frequency to occur than shorter ones (particularly due to simple sequence repeat), and hence a lower-level probability of homoplasy than short indels (Jiang et al. 2020). So, longer indels are therefore a valid tool for analyses on ancient evolutionary relationships, which may undergo complex evolutionary histories. Actually, it is easy to understand that, the convergent mutations for “several continuous sites” is obviously less frequent than those for single nucleotide. We can simply assume that, for orthologous sequences from two species, the probability of convergent mutation for one single-nucleotide could be n , while that for five continuous sites would be n^5 . If $n=0.1$, then n^5 would be 0.00001. We can find indels more than 5 bp in length could effectively minimize effects of homoplasy, and thus reveal the real phylogenies for ancient diverged lineages. On the other hand, enough information was needed for population genomic analyses. If we only retain much longer indels (such as those > 10 bp or > 20 bp), effective information would be much less. The primary HHS test (the test for significant PIVs) was developed by Jiang et al. (2020), which used “relatively long indels (> 5 bp)” as the genetic markers. So, we remained use this type of data in our improved population-level HHS test, and further applied it on our other population genomic analyses (ABBA-BABA test and Hyde analysis) according to its such advantage.

Bapteste E, Philippe H. The potential value of indels as phylogenetic markers: position of trichomonads as a case study. *Molecular Biology and Evolution*. 2002. 19: 972–977.

Rokas A, Holland P. Rare genomic changes as a tool for phylogenetics. *Trends in Ecology and*

Evolution. 2000. 15: 454–459.

Jiang YF, Yuan ZW, ..., Liu CJ. Differentiating homoploid hybridization from ancestral subdivision in evaluating the origin of the D lineage in wheat. *New Phytologist*. 2020. 228: 409–414.

3 — Much more detail needs to be provided about the implementation of the ABBA-BABA test and HyDe. First, it is unclear in Section 5.1 what the role of the estimated phylogeny is. Neither HyDe nor the ABBA-BABA test require this. What were the bootstrap support values used for? Why were the indels transformed into “pseudo-alleles”? HyDe handles full sequence data so this seems like a potential loss of information. What does “the analysis was performed at the population level” mean? Perhaps giving the specific HyDe commands used would help to clarify all of this. Similar comments apply for the ABBA-BABA test. Does the Dsuit software have options available? Which were used?

Response: Thanks for your good question. We estimated phylogeny based on population genomic data was aimed at determining the population-level phylogenetic relationships and examining whether they are identical with those produced by single genomes. Bootstrap values were used to assess the accuracy of phylogeny. It was a widely used method for phylogenetic analysis. To reduce the bias produced by detecting indels based on Illumina data and because recombination of each indel could be treated as an independent event, we discarded length information about the indels and set each of them to have the same weight. So, indels were transformed into “pseudo-alleles”. HyDe and ABBA-BABA test were performed at the population level, in which the information from all individuals per population were imputed. Their applications with population genomic data make them capable of detecting hybrid signals with genetic information from all offspring and can therefore distinguish HHS from introgression. We illustrated more details about above points. The commands for ABBA-BABA test and HyDe were also provided.

4 — The authors need to explain what they consider high support, and how support was measured.

Response: Thanks for your valuable suggestions. In the new submitted manuscript, we have made great efforts on - how to explain our conclusion was accurate with high support. We introduced more background of the research field, described more details of our methods, carefully illustrated our results, and made a full discussion on - why our manuscript was markedly strengthened in both significant discoveries and the ground-breaking novelty methods.

Minor typos/grammatical errors:

Response: Thanks for your scrupulous tips. We have corrected all of them as you suggested. The manuscript has also been edited by a native English speaker.

- line 4: “... has been increasingly recognized widely occurred”

Response: Corrected.

- line 6: change to: “focused on closely-related species”

Response: Corrected.

- line 6: “while it” — what does it refer to? This modifies “previous studies” so “it” doesn’t make sense

Response: Corrected.

- line 11 and throughout: why is “Indels” capitalized?

Response: Done. We have changed them to “indels”.

- line 13: “at utmost” — what does this mean? Please re-word (this occurs elsewhere in the manuscript, as well)

Response: Done.

- line 22 and line 24: “is” vs. “was” — watch tense

Response: Corrected.

- lines 22-23: change to “to distinguish”

Response: Corrected.

- line 26: re-word “difficulty to exclude homoplasy”

Response: Done.

- line 27, remove “the”

Response: Corrected.

- line 34, change “the” to “a”

Response: Corrected.

- line 43, change to “The genomes of these three species”

Response: Corrected.

- line 46, “were revealed” — re-word

Response: Done.

- line 47, “high-confidence supports” — this is unclear

Response: Corrected.

- lines 47 and 50: “the 2,414 ones” and “906 ones” — re-word

Response: Corrected.

- line 51, remove “the”

Response: Corrected.

- line 52: “the ILS hypothesis” — what does this mean? ILS is an evolutionary process. Please re-word. I think the authors mean the hypothesis that gene tree incongruence is due to solely to the process of ILS and not to hybridization.

Response: Corrected.

- line 70, the phrase, “Based on the [I]ndel spectrum” is not clear.

Response: Corrected.

- line 71: change to “used the ABBA-BABA test”

Response: Corrected.

- line 73, remove the words “following” and “the”

Response: Corrected.

- lines 80 and 81: “the most number” and “the second” — re-word

Response: Corrected.

- line 87, what “intermediate morphological trait” is being referred to here?

Response: Done. We have revised this sentence to state it clearly.

- line 100, remove “of”

Response: Corrected.

- line 107, change to “reveal a case of ancient intergeneric HHS”

Response: Corrected.

- line 178 (and elsewhere), the wording “to detect the likely hybridization event” is odd. The authors are actually testing to see whether the data support hybridization, right?

Response: You are right. We have revised this statement here and elsewhere.

- line 192, the phrase “the top 2.5% number of fixed non-synonymous mutation sites” is unclear

Response: We have revised this sentence.

- line 196, “in both grouping ways” — this is unclear

Response: We have revised this sentence.

- The title of Section 3.4 in the Supplementary Information is misleading. The authors have written, “The exclusion of the incomplete lineage sorting effects by simulation”. But I think what they mean is that ILS **alone** cannot explain the patterns observed in the gene tree topologies.

Response: Thanks for your helpful suggestion. We have revised this title.

Reviewers' Comments:

Reviewer #1:

Remarks to the Author:

This revised MS is clearer and more persuasive than the original submission that I reviewed rather critically some months ago. The MS remains somewhat controversial, but I think that its results are sound enough for debate over the author's findings to now be carried out in the literature, rather than in the peer review process. My main suggestion is that the authors tone it down a little, and express their conclusions slightly more tentatively than they do currently. I also suggest they start the MS with a reference to Lotsy, J.P. 1916. Evolution by means of hybridisation (The Hague: Martinus Nijhoff). In this book Johannes Lotsy suggests that hybridisation can drive evolution at higher taxonomic levels. This is directly relevant to the MS in hand, and will help set it in the context of the historical literature.

Reviewer #2:

Remarks to the Author:

This paper tests population genomic data for a taxonomically difficult group of tree species across three related genera and presents results arguing for an ancient hybrid origin of one of these genera. The aims of the paper have been revised with more accurate referencing as requested to test alternative hypotheses for the origin of the controversial genus *Distegocarpus* rather than confirming hybrid origin. Revised figure 2 is a useful addition to confirm the gene tree biases favouring hybridization hypotheses over ILS hypotheses. The arguments for distinguishing between introgression and hybrid speciation using population genomic data are convincing and supported by the extensive literature search that the authors provide. Redrawn Figure 3 supports also these conclusions. These hypotheses are now clearly explained within the results.

My queries about divergence times have been answered. Justification of the reference genome and tests of potential ascertainment bias have been presented. The authors developed the RI argument at my suggestion to include acknowledgement of continuing RI evolution since hybrid origin but to me this still seems the more speculative part of the story and could perhaps be toned down with more cautious language. There is still no direct testing of RI and therefore the authors should probably note that their interpretations remain to be confirmed as part of future research. There is no formal functional enrichment analysis of the PSGs that the authors identify so listing a few genes with functions of interest feels rather subjective. I would also suggest that the authors provide some extra explanation about why genes related to flowering and habitat and long indels could contribute to RI as I feel these arguments could be lost for less specialized readers.

The responses to other reviewers are thorough and improve the paper by strengthening the arguments for how the results support HHS over introgression, particularly the arguments based around population level testing and use of indel variants in the revised discussion.

I am satisfied that the paper has been improved by revision and presents strong evidence for its major conclusion that will be of wide interest to readers in the field of evolution.

Specific comments

L201 Replace "enforced" with "reinforced"

Reviewer #3:

Remarks to the Author:

Thank you for addressing my comments in detail. While I do not agree everything with their interpretations I do laud their effort in convincing me and potentially the field of their results. Peer review is not for me to decide whether this paper is right or wrong because I am N=1 of an opinion. I'll leave up to the field to decide.

I have no further comments and deem the paper suitable for publication.

Reviewer #4:

Remarks to the Author:

The revised version of the manuscript is significantly improved, both with regard to the writing style and with regard to the justification for HHS. I'm sure it is difficult to write in a language other than one's native language, and I appreciate the care the authors have taken with this in their revision. I'll reiterate that I think the authors have taken a thorough approach to address an interesting question and feel that the work will have high impact.

All of the comments from my previous review have been sufficiently addressed. I have just a few comments.

Major:

- At several places in the manuscript (e.g., lines 136 and 179), the authors say that they are "excluding" ILS (or similar wording). This is not accurate, and failing to adjust this wording will propagate this error in the literature. They are not excluding ILS as an explanation for incongruence, as ILS is a process that is known to occur across genes in genomic studies. Rather, they are deciding that ILS *alone* is not a sufficient explanation for the observed degree of gene tree incongruence. These are different, and the authors need to write about them as such.

HyDe, for example, is a test for hybridization *in the presence of lineage sorting*; in other words, the statistical work underlying the hypothesis test in HyDe assumes that ILS occurs. It then asks whether ILS alone is sufficient to explain the data, or whether hybridization in combination with ILS provides a better explanation.

I think this can mostly be remedied by some small wording changes:

- line 136, change to "exclude the ILS only .."
- line 179, change to "We excluded ILS alone as an explanation of observed patterns in the data based on an ABBA-BABA test (D-statistic) and HyDe analysis."

Minor:

- line 39, "This is supported" - perhaps be more specific about what "This" refers to.
- line 63, "With rapidly growing of genomics" -> "With the rapid growth of genomics"
- legend to Figure 2, "The grey bars indicate 10,000 times simulated ratios" -> "The grey bars are a histogram of the ratios observed in 10,000 simulated data sets."
- line 461, "used lineages" -> "lineages used"

Reviewer #5:

Remarks to the Author:

The manuscript NCOMMS-21-25629A-Z attempted to explain the homoploid hybrid speciation (HHS) of *Carpinus* sect. *Distegocarpus* by whole genome sequencing of *C. viminea* and re-sequencing of 47 individuals in combination with the previous reports of two genomes for *C. fangiana* and *Ostrya rehderiana* to prove that *C. distegocarpus* originates from introgression of genus *Carpinus* and genus *Ostrya*. Although the methods used in this manuscript seem applicable, the novelty is lacking. The authors provided evidence for the HHS for two species or the two species are originated from two different genera, but in fact the difference between intrageneric and interspecific species is limited, because the genes of these species are homologous based on phylogenetic tree (Fig. 1), and species might be classified to different genera. Even the sects. between *Distegocarpus* and *Carpinus* may from an excessive division, this is perhaps the key question for hybrid speciation research. Hybridization

and reticulate evolution are the main drivers for speciation in plants, but RAD-seq or low-coverage sequencing could be sufficient already to solve the problem. The genome assembly has a contig N50 value of 4.3 Mb which is acceptable while from a result of HiC, the assembly quality is deficient that might impact the collinearity analysis results, a better quality is therefore desired (Supplementary Figs. 3, 4). Although the genome assembly quality is sufficient to address the HSS, content including genome structure, characteristics and evolution was not presented. Overall, the scientific questions in this manuscript do not raise significant interest to the genomic community, the substantial work only provides evidence for HSS which is more like a section of a genome paper, more analyses concerning in-depth questions should be conducted to advance the manuscript. It's suggested to address the question of species taxonomy and adopt new sequencing techniques to improve assembly quality and make a focus to a genome evolution story.

REVIEWER COMMENTS

Reviewer #1 (Remarks to the Author):

This revised MS is clearer and more persuasive than the original submission that I reviewed rather critically some months ago. The MS remains somewhat controversial, but I think that its results are sound enough for debate over the author's findings to now be carried out in the literature, rather than in the peer review process. My main suggestion is that the authors tone it down a little, and express their conclusions slightly more tentatively than they do currently. I also suggest they start the MS with a reference to Lotsy, J.P. 1916. Evolution by means of hybridisation (The Hague: Martinus Nijhoff). In this book Johannes Lotsy suggests that hybridisation can drive evolution at higher taxonomic levels. This is directly relevant to the MS in hand, and will help set it in the context of the historical literature.

Reply: Thanks for your positive comments and great suggestions, helping improve our manuscript greatly. We have toned down the expression of our conclusions in the revised manuscript, in the Abstract, Results and Discussion: we deleted all conclusive claims and for all conclusions, we added the 'likely' and 'possible' worlds. We have also cited the above reference and started the manuscript following it according to your suggestions (L22-L23). We hope our manuscript is suitable for publication now.

Reviewer #2 (Remarks to the Author):

This paper tests population genomic data for a taxonomically difficult group of tree species across three related genera and presents results arguing for an ancient hybrid origin of one of these genera. The aims of the paper have been revised with more accurate referencing as requested to test alternative hypotheses for the origin of the controversial genus *Distegocarpus* rather than confirming hybrid origin. Revised figure 2 is a useful addition to confirm the gene tree biases favouring hybridization hypotheses over ILS hypotheses. The arguments for distinguishing between introgression and hybrid speciation using population genomic data are convincing and supported by the extensive literature search that the authors provide. Redrawn Figure 3 supports also these conclusions. These hypotheses are now clearly explained within the results.

Reply: Thanks for your positive comments.

My queries about divergence times have been answered. Justification of the reference genome and tests of potential ascertainment bias have been presented. The authors developed the RI argument at my suggestion to include acknowledgement of continuing RI evolution since hybrid origin but to me this still seems the more speculative part of the story and could perhaps be toned down with more cautious language. There is still no direct testing of RI and therefore the authors should probably note that their interpretations remain to be confirmed as part of future research. There is no formal functional enrichment analysis of the PSGs that the authors identify so listing a few genes with functions of interest feels rather subjective. I would also suggest that the authors provide some extra explanation about why genes related to flowering and habitat and long indels could contribute to RI as I feel these arguments could be lost for less specialized readers.

Reply: Thanks for your careful reviewing and valuable suggestions. We have further toned down the argument of RI with more cautious language (e.g., L205, L210, L221 et al.) and especially pointed that the

further functional tests of these alleles related are highly needed in the future (L211-L212). We also toned down all claimed in the abstract and discussion. We also have provided the extra explanation for RI-related PSGs we selected and listed (such as those related to flowering and habitat, L172 and L206) and for the contributions of long indels to RI (L214). We believe these perfect suggestions will help to highly improve our manuscript.

The responses to other reviewers are thorough and improve the paper by strengthening the arguments for how the results support HHS over introgression, particularly the arguments based around population level testing and use of indel variants in the revised discussion.

I am satisfied that the paper has been improved by revision and presents strong evidence for its major conclusion that will be of wide interest to readers in the field of evolution.

Reply: Thanks again for your positive comments and valuable suggestions. We have further revised our manuscript according to your suggestion. We hope it is more suitable for publication now.

Specific comments

L201 Replace "enforced" with "reinforced"

Reply: Done. Thanks for your carefulness.

Reviewer #3 (Remarks to the Author):

Thank you for addressing my comments in detail. While I do not agree everything with their interpretations I do laud their effort in convincing me and potentially the field of their results. Peer review is not for me to decide whether this paper is right or wrong because I am N=1 of an opinion. I'll leave up to the field to decide. I have no further comments and deem the paper suitable for publication.

Reply: Thanks for your positive comments. We really appreciate for your constructive and careful reviewing last time, which help us highly improve our manuscript.

Reviewer #4 (Remarks to the Author):

The revised version of the manuscript is significantly improved, both with regard to the writing style and with regard to the justification for HHS. I'm sure it is difficult to write in a language other than one's native language, and I appreciate the care the authors have taken with this in their revision. I'll reiterate that I think the authors have taken a thorough approach to address an interesting question and feel that the work will have high impact.

Reply: Thanks for your positive comments.

All of the comments from my previous review have been sufficiently addressed. I have just a few comments.

Major:

- At several places in the manuscript (e.g., lines 136 and 179), the authors say that they are "excluding" ILS (or

similar wording). This is not accurate, and failing to adjust this wording will propagate this error in the literature. They are not excluding ILS as an explanation for incongruence, as ILS is a process that is known to occur across genes in genomic studies. Rather, they are deciding that ILS *alone* is not a sufficient explanation for the observed degree of gene tree incongruence. These are different, and the authors need to write about them as such.

HyDe, for example, is a test for hybridization *in the presence of lineage sorting*; in other words, the statistical work underlying the hypothesis test in HyDe assumes that ILS occurs. It then asks whether ILS alone is sufficient to explain the data, or whether hybridization in combination with ILS provides a better explanation.

I think this can mostly be remedied by some small wording changes:

- line 136, change to "exclude the ILS only .."

- line 179, change to "We excluded ILS alone as an explanation of observed patterns in the data based on an ABBA-BABA test (D-statistic) and HyDe analysis."

Reply: Thanks very much. We appreciate for your valuable suggestions. We are so sorry for not correcting all of such statements last time. Now, we have corrected them according to your suggestions.

Minor:

- line 39, "This is supported" - perhaps be more specific about what "This" refers to.

Reply: Thanks very much. We have revised it according to your suggestion.

- line 63, "With rapidly growing of genomics" -> "With the rapid growth of genomics"

Reply: Corrected. Thanks very much.

- legend to Figure 2, "The grey bars indicate 10,000 times simulated ratios" -> "The gray bars are a histogram of the ratios observed in 10,000 simulated data sets."

Reply: Corrected. Thanks very much.

- line 461, "used lineages" -> "lineages used"

Reply: Corrected. Thanks very much.

Reviewer #5 (Remarks to the Author):

The manuscript NCOMMS-21-25629A-Z attempted to explain the homoploid hybrid speciation (HHS) of *Carpinus* sect. *Distegocarpus* by whole genome sequencing of *C. viminea* and re-sequencing of 47 individuals in combination with the previous reports of two genomes for *C. fangiana* and *Ostrya rehderiana* to prove that *C. distegocarpus* originates from introgression of genus *Carpinus* and genus *Ostrya*. Although the methods used in this manuscript seem applicable, the novelty is lacking. The authors provided evidence for the HHS for two species or the two species are originated from two different genera, but in fact the difference between intrageneric and interspecific species is limited, because the genes of these species are homologous based on phylogenetic tree (Fig. 1), and

species might be classified to different genera. Even the sects. between *Distegocarpus* and *Carpinus* may from an excessive division, this is perhaps the key question for hybrid speciation research. Hybridization and reticulate evolution are the main drivers for speciation in plants, but RAD-seq or low-coverage sequencing could be sufficient already to solve the problem.

The genome assembly has a contig N50 value of 4.3 Mb which is acceptable while from a result of HiC, the assembly quality is deficient that might impact the collinearity analysis results, a better quality is therefore desire (Supplementary Figs. 3, 4). Although the genome assembly quality is sufficient to address the HHS, content including genome structure, characteristics and evolution was not presenting. Overall, the scientific questions in this manuscript do not raise significant interest to the genomic community, the substantial work only provides evidence for HSS which is more like a section of a genome paper, more analyses concerning in-depth questions should be conducted to advance the manuscript. It's suggested to address the question of species taxonomy and adopt new sequencing technique to improve assembly quality and make a focus to a genome evolution story.

Reply: Thanks very much for your valuable comments. Actually, our study (the speciation research) is mainly emphasis on the fields of ecology and evolution, but not solely the genome evolution. As you pointed out, hybridization and hybrid speciation contribute greatly to reticulate evolution and diverse biodiversity in the world. This may cause widespread phylogenetic inconsistencies in constructing tree life (Soltis and Soltis 2019, Nature Plants) and even affect our understanding of the basic 'species concept' (Science 309, 78–102, 2005). Since Darwin, speciation mechanism has been one of the important topics that is of high interest to many researchers, the total science community, and the general public.

Previous studies on homoploid hybrid speciation (HHS) have mostly focused on the closely related species (e.g., Mavárez et al. 2006, Nature; Lamichhaney et al. 2018, Science; Zhang et al. 2019, Molecular Biology and Evolution; Wang et al. 2021, Molecular Plant); however, it has been rarely reported or detected between ancestors of different genera (Jiang et al. 2020, New Phytologist). The HHS occurrences between genera and species is significant for our understanding of tree of life and the origin of the high and low taxonomic entity. A major challenge in identifying a rather ancient HHS event (like such HHS between different genera in our research) is the difficulty to exclude homoplasmy (Jiang et al. 2020, New Phytologist), which may derive from random and/or convergent nucleotide mutations during the long-term evolutionary histories (Rokas and Holland 2000, Trends in Ecology and Evolution; Baptiste and Philippe 2002, Molecular Biology and Evolution). Thus, the traditional methods based on homologs or SNPs are usually noneffective for the deep-diverged lineages. In this study, we first applied and highlighted the importance of long indels with least evolutionary homoplasmy at the population level to identify ancient HHS events. In consequence, for indels detecting, high-coverage whole-genome sequencing is necessary. This is a great innovation of methodology in HHS research.

Herein, we reported an ancient HHS case between different genera. After the origin, the hybrid lineage and its two parents further produce multiple daughter species. Such an initial HHS event before further species diversification may lead to inconsistent gene tree topologies and reticulate phylogenies widely reported in plants at higher taxonomic levels. To our knowledge, this is the first intergeneric HHS case with the most ancient time scale. We believe that our study will attract those researchers to detect the ancient HHS events in their focused groups, and will be cited frequently by the numerous research groups in the world. These reasons will bring in a broad readership.

We appreciate for your valuable suggestions on genome assembly and evolutionary analyses. We reanalyzed

and improved the Hi-C assembly for *Ostrya rehderiana* (Fig. S4) and thus further improved the results of collinearity analysis (Fig. S5). We also performed the analyses of genome structure, characteristics, and evolution according to your suggestions, and presented them in a new Fig. 1 and added the description of related results (L82-L91) and methods (L259-L271 in the Main Text and a new Supplementary Note 3 in the Supplementary Information) in the manuscript. However, considering the long-time cost (especially for the COVID-19 pandemic), it is really difficult for us to adapt new sequencing technique to improve the assembly quality in this period. We agree with you that, the current genome assembly quality is sufficient to address the HHS. So, it is maybe better to generate a further improved assembly by new sequencing technique in the future research.

We believe we have addressed an interesting question in the fields of ecology and evolution, and our work will have a high impact. We really hope we can convince you.

References:

- Baptiste E. & Philippe H. The potential value of indels as phylogenetic markers: position of trichomonads as a case study. *Molecular Biology and Evolution* 19, 972–977 (2002).
- Jiang Y. F. et al. Differentiating homoploid hybridization from ancestral subdivision in evaluating the origin of the D lineage in wheat. *New Phytologist* 228, 409–414 (2020).
- Lamichhaney S. et al. Rapid hybrid speciation in Darwin’s finches. *Science* 359, 224–228 (2018).
- Mavárez J. et al. Speciation by hybridization in *Heliconius* butterflies. *Nature* 441, 868–871 (2006).
- Rokas A. & Holland P. W. Rare genomic changes as a tool for phylogenetics. *Trends in Ecology and Evolution* 15, 454–459 (2000).
- Soltis D. E. & Soltis P. S. Nuclear genomes of two magnoliids. *Nature Plants* 5, 6–7 (2019).
- Wang Z. F. et al. Hybrid speciation via inheritance of alternate alleles of parental isolating genes. *Molecular Plant* 14, 208–222 (2021).
- Zhang B. W. et al. Phylogenomics reveals an ancient hybrid origin of the Persian walnut. *Molecular Biology and Evolution* 36, 2451–2461 (2019).

Reviewers' Comments:

Reviewer #1:

Remarks to the Author:

I am now content that this study is worthy of publication

Reviewer #2:

Remarks to the Author:

I thank the authors for acting upon my additional suggestions to add detail and explanation to the RI section. The authors have also provided new synteny analysis to provide further context to their study. I am satisfied to recommend this paper.

Reviewer #4:

Remarks to the Author:

The authors have responded well to all of my concerns. I believe that the manuscript is ready for publication. The manuscript makes an important contribution to the literature in this area.

Reviewer #5:

Remarks to the Author:

The authors have respond to all my concerns, I have no further questions.